# c-di-GMP inhibits the DNA binding activity of H-NS in *Salmonella*

Shuyu Li[1], Qinmeng Liu[1], Chongyi Duan[1], Jialin Li[1], Hengxi Sun[1], Lei Xu[1], Qiao Yang[2,3], Yao Wang[1], Xihui Shen[1] ✉ & Lei Zhang[1] ✉

Cyclic di-GMP (c-di-GMP) is a second messenger that transduces extracellular stimuli into cellular responses and regulates various biological processes in bacteria. H-NS is a global regulatory protein that represses expression of many genes, but how H-NS activity is modulated by environmental signals remains largely unclear. Here, we show that high intracellular c-di-GMP levels, induced by environmental cues, relieve H-NS-mediated transcriptional silencing in *Salmonella enterica* serovar Typhimurium. We find that c-di-GMP binds to the H-NS protein to inhibit its binding to DNA, thus derepressing genes silenced by H-NS. However, c-di-GMP is unable to displace H-NS from DNA. In addition, a K107A mutation in H-NS abolishes response to c-di-GMP but leaves its DNA binding activity unaffected in vivo. Our results thus suggest a mechanism by which H-NS acts as an environment-sensing regulator in Gram-negative bacteria.

Cyclic di-GMP (c-di-GMP) is an important bacterial second messenger, with its synthesis and degradation catalyzed by diguanylate cyclases (DGCs) and c-di-GMP-specific phosphodiesterases (PDEs), respectively[1]. Bacteria use c-di-GMP to control a wide range of biological processes, including motility, biofilm formation, cell cycle progression and virulence[1]. While extracellular stimuli can modulate intracellular levels of c-di-GMP by regulating the expression or activity of DGCs or PDEs, c-di-GMP-binding effectors convert dynamic changes in c-di-GMP concentration into specific cellular responses[2]. c-di-GMP effectors identified include mRNA riboswitches, transcription factors, a broad variety of enzymes, adapter proteins, and chaperone proteins[1–4]. Nevertheless, the mechanisms involved in c-di-GMP metabolism and recognition are not yet fully understood.

Histone-like nucleoid structuring protein (H-NS) is a nucleoid-associated protein participating in nucleoid organization and transcription regulation in Gram-negative bacteria[5,6]. H-NS acts primarily as a transcription silencer, repressing the expression of hundreds of genes throughout the bacterial genome[5–8]. H-NS has been shown to selectively silence xenogeneic DNA regions that are derived from a foreign source and possess AT content typically higher than that of the resident genome, which are usually associated with virulence and adaptive stress responses[5,7–9]. Xenogeneic silencing by H-NS is thought to prevent the inappropriate expression of horizontally acquired genes and thus suppress their detrimental effect on bacterial fitness[10,11]. On the other hand, bacteria have evolved various mechanisms to derepress H-NS-silenced genes in order to benefit from their expression in specific circumstances[12]. For instance, H-NS has been shown to act as an environmental sensor, with changes in temperature and osmolarity affecting the DNA binding properties of H-NS and thus activating H-NS-repressed genes such as the virulence gene *virF* in *Shigella flexneri* and the type VI secretion system (T6SS) gene *hcp1* in *Vibrio parahaemolyticus*[5,13]. However, environmental signals that modulate the DNA binding activity of H-NS and the underlying mechanisms are still largely unknown.

In this work, we show that elevated c-di-GMP levels induced by environmental and host-derived cues such as L-arginine and bile salts promote the transcription of H-NS-repressed T6SS genes in *Salmonella enterica* serovar Typhimurium. We further find that H-NS is a c-di-GMP-responsive transcriptional regulator. Binding of c-di-GMP to H-NS abrogates the binding of H-NS to DNA, thus relieving H-NS-imposed gene silencing. Our study reveals a previously unrecognized anti-H-NS mechanism mediated by c-di-GMP and greatly expands the range of environmental signals to which H-NS-silenced genes respond.

[1]State Key Laboratory of Crop Stress Biology for Arid Areas, Shaanxi Key Laboratory of Agricultural and Environmental Microbiology, College of Life Sciences, Northwest A&F University, Yangling, Shaanxi 712100, China. [2]ABI Group, College of Marine Science and Technology, Zhejiang Ocean University, Zhoushan 316021, China. [3]Donghai Laboratory, Zhoushan 316021, China. ✉e-mail: xihuishen@nwsuaf.edu.cn; zhanglei0075@nwsuaf.edu.cn

## Results

### High levels of c-di-GMP upregulate the transcription of *S.* Typhimurium T6SS genes via an H-NS-dependent regulatory pathway

The activity of T6SSs from important intestinal pathogens such as *Vibrio cholerae* and *S.* Typhimurium has been shown to be activated by bile salts[14,15], but the mechanism of such regulation is still elusive. Several studies indicate that c-di-GMP signaling regulates T6SS activity in pathogenic bacteria such as *P. aeruginosa* and *Agrobacterium tumefaciens*[16,17], while our recent study demonstrated that bile salts induce intracellular accumulation of c-di-GMP in *S.* Typhimurium[4], leading us to speculate that bile salts activate *S.* Typhimurium SPI-6-

encoded T6SS via modulating intracellular c-di-GMP levels. While stimulating a ~3-fold increase in intracellular c-di-GMP concentrations via the DGC YedQ[4] (Supplementary Fig. 1a), bile salts promoted Hcp1 secretion in the parental strain, but not in the Δ*yedQ* mutant (Fig. 1a). Increased Hcp1 secretion in response to bile salts was restored when the mutant was complemented with a plasmid-derived copy of *yedQ* (Fig. 1a). L-Arginine, an environmental signal that increases the intracellular c-di-GMP concentration in *S.* Typhimurium[18] (Supplementary Fig. 1b), also promoted Hcp1 secretion in the wild-type background (Supplementary Fig. 2). Furthermore, an increase in cellular levels of c-di-GMP by overexpression of *adrA* (*STM0385*) that encodes a DGC (Supplementary Fig. 1c) promoted the secretion of Hcp1, whereas a

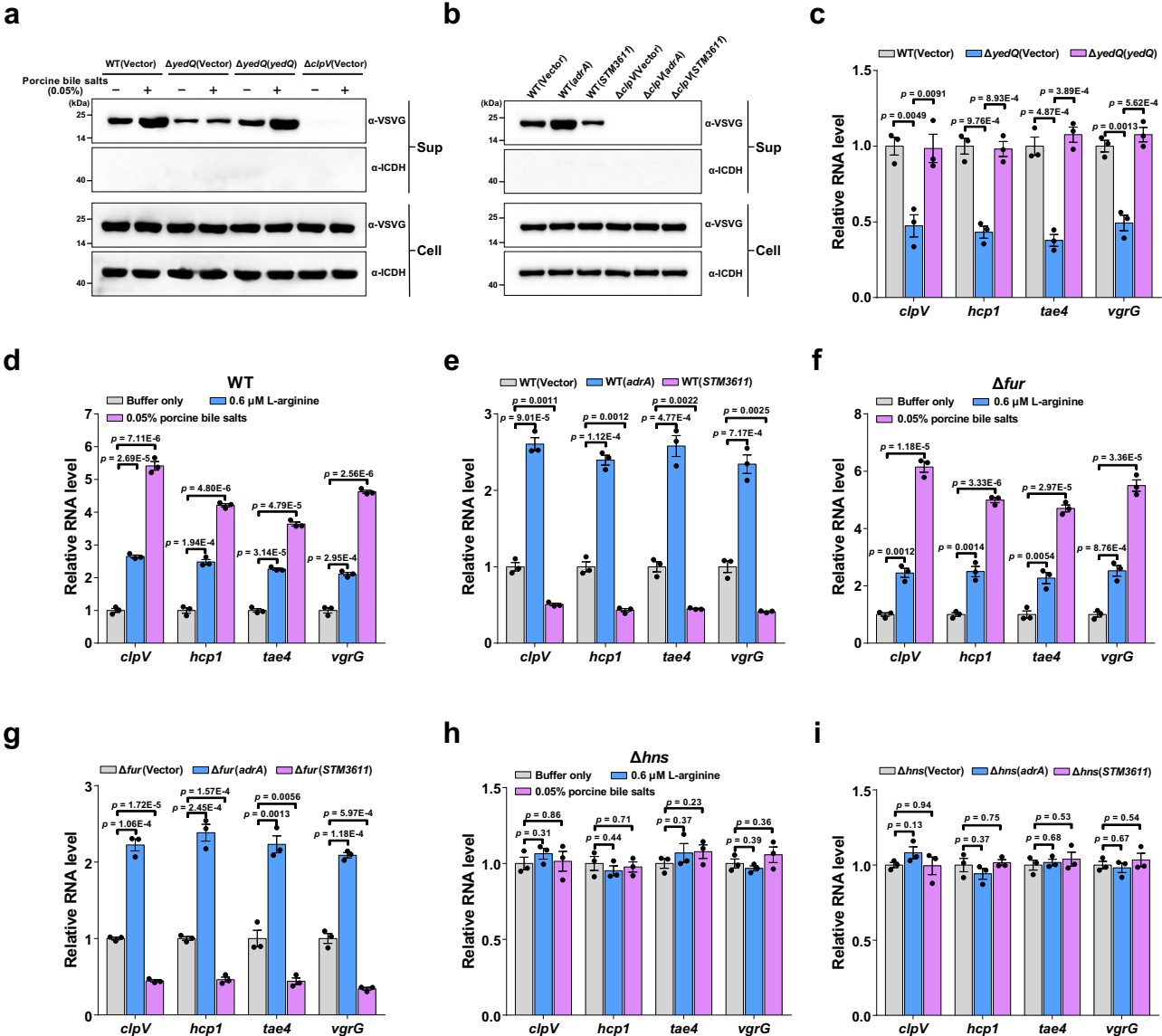

**Fig. 1 | Elevated c-di-GMP levels upregulate the transcription of *S.* Typhimurium T6SS genes via an H-NS-dependent pathway. a** The activity of the SPI-6 T6SS is enhanced by bile salts in the wild-type (WT) strain, but not in Δ*yedQ*. Western blot analysis of Hcp1 with a C-terminal vesicular stomatitis virus glycoprotein (VSVG) tag was performed in cell pellet (Cell) and concentrated supernatant (Sup) from *S.* Typhimurium strains carrying a plasmid expressing Hcp1-VSVG. An antibody against isocitrate dehydrogenase (ICDH) is used as a loading control. **b** The T6SS activity of WT *S.* Typhimurium is modulated by overexpression of *adrA* or *STM3611*. **c** qRT-PCR analysis of T6SS gene expression in the WT, Δ*yedQ* mutant and complemented strains. **d, e** qRT-PCR analysis of T6SS gene expression in WT *S.* Typhimurium stimulated by L-arginine or bile salts (**d**) or its derivatives

overexpressing *adrA* or *STM3611* (**e**). **f, g** qRT-PCR analysis of T6SS gene expression in Δ*fur* stimulated by L-arginine or bile salts (**f**) or its derivatives overexpressing *adrA* or *STM3611* (**g**). **h, i** qRT-PCR analysis of T6SS gene expression in Δ*hns* stimulated by L-arginine or bile salts (**h**) or its derivatives overexpressing *adrA* or *STM3611* (**i**). **a, b** Blots shown are representative of three independent experiments with similar results. **c–i** Gene expression levels were reported as fold change relative to that of the WT (**c–e**), Δ*fur* (**f, g**) or Δ*hns* (**h, i**) without stimulation or gene overexpression. Data are mean ± SD of three biological replicates. Two-sided, unpaired Student's *t*-test was used for statistical analyses, and $p < 0.05$ were considered to indicate statistically significant differences. Source data are provided as a Source Data file.

decrease in intracellular c-di-GMP concentrations by overexpression of *STM3611* that encodes a PDE (Supplementary Fig. 1c) reduced its secretion (Fig. 1b). These data indicate that high levels of c-di-GMP activate the T6SS in *S.* Typhimurium.

Quantitative real-time PCR (qRT-PCR) analysis showed that the mRNA levels of T6SS genes including *clpV*, *hcp1*, *tae4* and *vgrG* were significantly lower in the Δ*yedQ* mutant compared to the wild-type and complemented strains (Fig. 1c). In contrast, the mRNA levels of the four T6SS genes in the wild type were ~2–5-fold increased after stimulation with L-arginine or bile salts (Fig. 1d). Moreover, *adrA* overexpression in the wild type upregulated, whereas *STM3611* overexpression down-regulated, the mRNA levels of the four T6SS genes by more than 2-fold (Fig. 1e). Consistently, promoter-reporter assays showed that the promoter activities of *clpV*, *hcp1*, *tae4* and *vgrG* in the wild type were significantly increased following the addition of L-arginine or bile salts (Supplementary Fig. 3a) or the overexpression of *adrA*, but were significantly inhibited following the overexpression of *STM3611* (Supplementary Fig. 3b). Collectively, these results indicate that c-di-GMP positively regulates the expression of T6SS genes at the transcriptional level.

Fur was shown to repress the expression of *clpV* by binding directly to the *clpV* promoter, and might be able to indirectly regulate T6SS expression by repressing transcription of the T6SS repressor H-NS[19,20]. While the deletion of *fur* led to increased expression of *clpV* (Supplementary Fig. 4a), L-arginine and bile salts still induced expression of *clpV*, *hcp1*, *tae4* and *vgrG* in Δ*fur* (Fig. 1f). Moreover, overexpression of *adrA* and *STM3611* significantly upregulated and downregulated, respectively, the mRNA levels of the four T6SS genes in Δ*fur* (Fig. 1g). These results suggest that elevated c-di-GMP levels activate the T6SS gene expression in a Fur-independent manner.

Despite the fact that deletion of *hns* is lethal in some *S.* Typhimurium strains[5,7,21], an *hns* deletion mutation was successfully constructed by the CRISPR-Cas9 system in strain SL1344. The mutant exhibited only slightly retarded growth compared with the wild-type and complemented strains (Supplementary Fig. 5). Although deletion of *hns* tends to result in compensatory lesions in *stpA*, *rpoS*, or the *phoPQ* operon[7,22–24], no mutations in these genes were detected in the Δ*hns* mutant. However, mRNA level of *stpA* was upregulated more than 2-fold, whereas those of *rpoS* and the *phoPQ* operon were not altered in Δ*hns* compared to the wild type (Supplementary Fig. 6). Consistent with previous studies[7,21], deletion of *hns* led to significantly increased expression of *clpV*, *hcp1*, *tae4* and *vgrG* (Supplementary Fig. 4b). By contrast, although stimulation with L-arginine or bile salts, as well as overexpression of *adrA* or *STM3611* resulted in similar changes in intracellular c-di-GMP concentration in Δ*hns* compared to the wild type (Supplementary Fig. 1a–e), neither the stimulation nor the overexpression altered expression levels of the four T6SS genes in Δ*hns* (Fig. 1h, i). These results suggest that elevated c-di-GMP levels likely upregulate T6SS gene expression through an H-NS-dependent regulatory pathway.

### H-NS directly and specifically binds to c-di-GMP

Based on the above observations, we speculated that changes in intracellular c-di-GMP concentration may affect the production or activity of H-NS. We first generated *S.* Typhimurium strains in which a FLAG tag was fused to the N terminus of the native *hns* gene in the chromosome. Quantitative western blot analysis showed that overexpression of *adrA* or *STM3611* did not alter the production of H-NS (Supplementary Fig. 7), raising another possibility that H-NS is a c-di-GMP effector that regulates gene expression in response to this second messenger. We first tested the direct interaction between H-NS and the biotinylated c-di-GMP by an ultraviolet (UV)-crosslinking assay. YcgR, as a positive control, could clearly bind biotinylated c-di-GMP, while a negative control employing InvF did not show any biotin-binding signal (Fig. 2a). Intriguingly, under the same experimental conditions, we

detected a strong biotin-binding signal corresponding to H-NS bound to biotinylated c-di-GMP (Fig. 2a). Furthermore, the addition of unlabeled c-di-GMP competitively inhibited the binding of biotinylated c-di-GMP to H-NS, whereas unlabeled c-di-AMP and cGMP failed to impair biotinylated c-di-GMP binding (Fig. 2a), suggesting that H-NS can directly and specifically binds c-di-GMP. Furthermore, binding analysis by isothermal titration calorimetry (ITC) showed that c-di-GMP bind to H-NS with a binding affinity ($K_d$) of $0.27 \pm 0.04\,\mu M$ and 1:1 stoichiometry (Fig. 2b). The $K_d$ value presented here is comparable to those previously reported for other well-established c-di-GMP receptors[3,4,25]. By contrast, interactions of H-NS with c-di-AMP or cGMP were not detected under the same experimental conditions (Supplementary Fig. 8a, b). These results confirmed the binding specificity of H-NS for c-di-GMP.

The C-terminal DNA-binding domain (H-NS$_{Ctd}$, residues 91–137) of H-NS was found to show similar binding affinity to c-di-GMP compared to the full-length H-NS (Fig. 2b, c), suggesting that c-di-GMP binds to H-NS$_{Ctd}$ but not to the N-terminal domain of H-NS. Potential ligand-binding sites of the solution structure of H-NS$_{Ctd}$ (PDB ID: 2L93; https://www.rcsb.org/structure/2L93)[9] were predicted using POCASA 1.1[26] and molecular docking analysis by AutoDock Vina 1.1.2[27] showed that the best conformation of the H-NS$_{Ctd}$-c-di-GMP complex (Fig. 2d) has the lowest docking score of $-7.0\,kcal\,mol^{-1}$, which is below the threshold value of $-6\,kcal\,mol^{-1}$ for considering effective protein-ligand binding[28]. This conformation suggests that c-di-GMP makes close contact with Y99, D101, K107, R114, T115, P116, A117, V118 and K121 of H-NS$_{Ctd}$ (Fig. 2e). While purified H-NS variants with alanine substitution mutations in Y99, D101, K107 or T115 displayed similar size exclusion chromatography (SEC) elution profiles as wild-type H-NS (Supplementary Fig. 9), all the variants showed 63–178-fold reduced binding affinity for c-di-GMP (Fig. 2f), indicating that the four residues are important for the specific interaction between H-NS and c-di-GMP.

StpA is a closely-related paralogue of H-NS[10,29] and these two proteins share 51.8% sequence identity. Amino acid sequence alignment showed that StpA contains three of the four key residues in H-NS that are important for c-di-GMP binding (Supplementary Fig. 10a). In contrast to H-NS, StpA showed very low binding affinity ($K_d = 73 \pm 14\,\mu M$) for c-di-GMP (Supplementary Fig. 10b). Nevertheless, mutation of the non-conserved residue to conserved residue within StpA (F98Y) increased its c-di-GMP binding affinity to a level ($0.64 \pm 0.09\,\mu M$) (Supplementary Fig. 10b) comparable to that of H-NS (Fig. 2b). As the intracellular c-di-GMP concentrations in *S.* Typhimurium have been shown to vary from tens of nanomolar to low micromolar levels[18,30], the $K_d$ values presented here suggest that c-di-GMP is a physiologically relevant ligand for H-NS, but not StpA.

### c-di-GMP interferes with the binding of H-NS to DNA

Chromatin immunoprecipitation (ChIP) of in vivo cross-linked H-NS-DNA complexes followed by microarray (ChIP-on-chip) analysis has identified genomic loci bound by H-NS[7,21,31]. Consistent with these studies, our electrophoretic mobility shift assays (EMSAs) showed that H-NS specifically binds to the promoter sequences of the T6SS genes *clpV* and *vgrG*, as well as two other previously reported H-NS-repressed genes *fimA* and *yaiU* (Supplementary Fig. 11a–d). As expected, when c-di-GMP was added simultaneously with H-NS, this second messenger decreased the formation of the H-NS-DNA complexes in a concentration-dependent manner, whereas c-di-AMP and cGMP were unable to impair the H-NS-DNA complex formation (Fig. 3a–d). Moreover, binding analysis by ITC showed that H-NS binds the four promoter sequences with $K_d$ values of 0.29–0.46 $\mu M$, whereas the addition of c-di-GMP completely abolished the interactions of H-NS with its target promoters (Fig. 3e). However, when competitive EMSAs were performed by first incubating H-NS with the target promoters and then adding different concentrations of c-di-GMP, H-NS-DNA complexes were not dissociated by the subsequent addition of c-di-

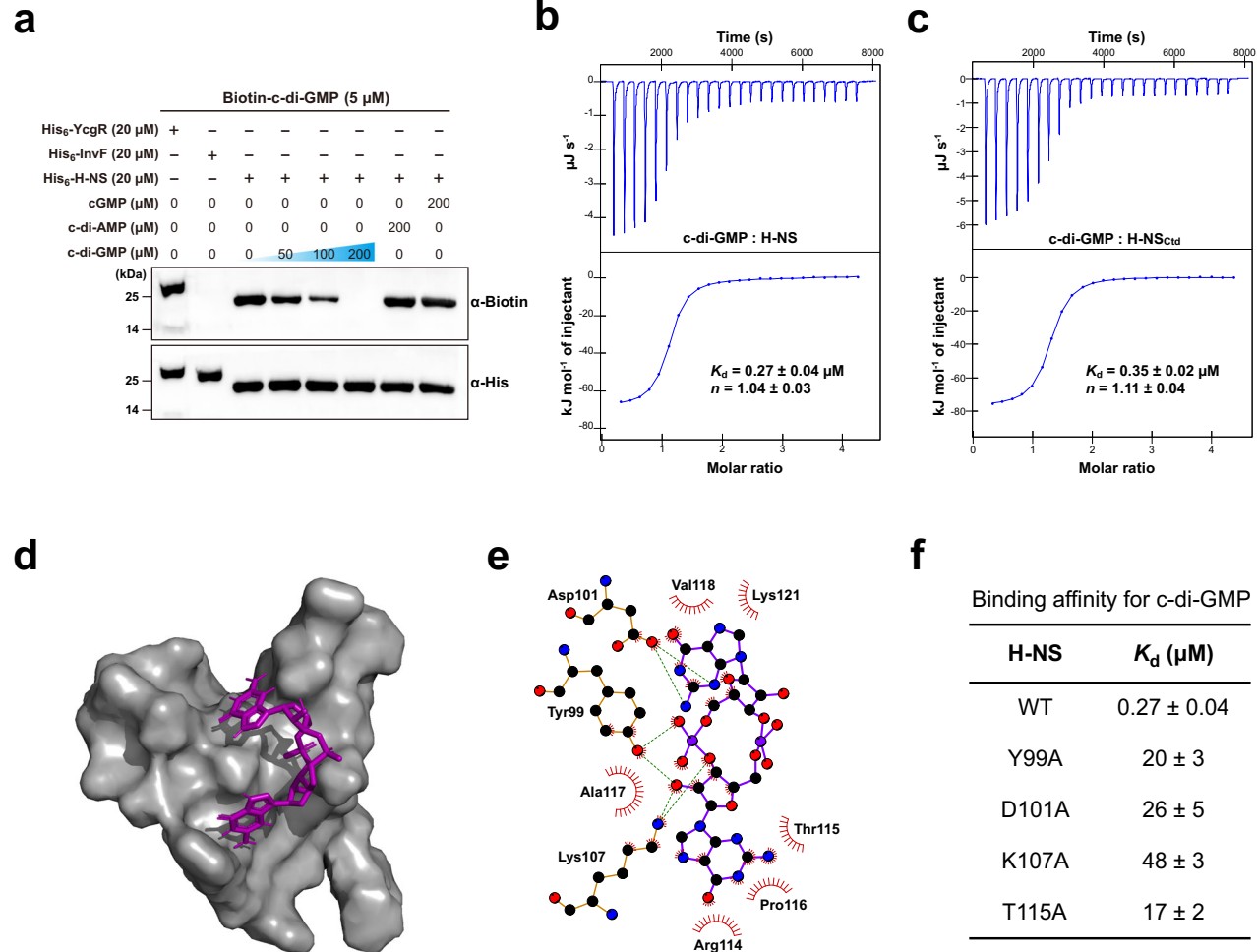

**Fig. 2 | Assays for specific interaction between H-NS and c-di-GMP. a** The UV-crosslinking assay detecting binding of His$_6$-H-NS to biotinylated c-di-GMP. His$_6$-YcgR was used as a positive control while His$_6$-InvF was employed as a negative control. A competitive experiment was performed by the addition of unlabeled c-di-GMP, c-di-AMP or cGMP to the reaction mixtures. Reaction samples were resolved by SDS-PAGE, transferred onto the membrane and probed with streptavidin-horseradish peroxidase (anti-biotin antibody) (top). The amounts of each protein that had been transferred to the membrane were also determined by western blot with the anti-His antibody (bottom). The blots shown are representative of three independent experiments with similar results. **b**, **c** c-di-GMP binds to H-NS (**b**) and H-NS$_{Ctd}$ (**c**) with high affinity. Data shown are one representative of three independent experiments with similar results, with $K_d$ and complex stoichiometry (*n*) presented as mean ± SD. **d** Surface representation of the structural model of H-NS$_{Ctd}$ in complex with c-di-GMP. c-di-GMP is shown as purple sticks. **e** Schematic of the predicted contacts between c-di-GMP and H-NS$_{Ctd}$. Potential hydrogen bonds are indicated as green dashed lines. **f** Binding of c-di-GMP to wild-type (WT) H-NS and its mutants. The binding affinity was measured by ITC. The $K_d$ values are presented as mean ± SD of three independent experiments. Source data are provided as a Source Data file.

GMP (Supplementary Fig. 12a–d). These results suggest that c-di-GMP inhibits DNA binding activity of H-NS but has no ability to displace H-NS from DNA.

To assess the effects of c-di-GMP on H-NS binding to DNA in cells, ChIP experiments were carried out with anti-FLAG antibody by using *S.* Typhimurium strains expressing in situ tagged FLAG-H-NS. FLAG-H-NS showed similar levels of c-di-GMP- and DNA-binding activity as the tag-free H-NS (Figs. 2b, 3e and Supplementary Fig. 13a), and the mRNA levels of the four H-NS-repressed genes were confirmed to be equivalent in FLAG-tagged and untagged wild-type *S.* Typhimurium (Supplementary Fig. 14), indicating that the function of H-NS was not impaired by the presence of the FLAG tag. ChIP-quantitative PCR (ChIP-qPCR) data showed that stimulation with L-arginine or bile salts and overexpression of *adrA* significantly reduced the enrichment of FLAG-H-NS at the promoters of the four genes, while overexpression of *STM3611* significantly increased its enrichment at these promoters (Fig. 3f, g). These results indicate that binding of c-di-GMP to H-NS inhibits the DNA binding activity of H-NS, thus promoting expression of H-NS-repressed genes.

## The K107A variant of H-NS maintains unaffected DNA binding activity but loses response to c-di-GMP in vivo

H-NS has been shown to bind DNA minor grooves through an AT-hook-like short loop containing the "Gln112-Gly113-Arg114" motif[9]. Among residues of H-NS that make contact with c-di-GMP (Fig. 2e), R114 and T115 were predicted to participate in interactions with DNA[9]. As expected, the mutation of T115 to alanine dramatically reduced its binding affinity for the promoter sequences of *clpV* and *vgrG* (Supplementary Fig. 13b). These results indicate that the interaction surface of H-NS with DNA partially overlaps with the c-di-GMP-binding site, which provides a mechanistic explanation for why c-di-GMP binding inhibits the binding of H-NS to DNA. While mutation of K107 led to a drastic reduction in the binding affinity of H-NS for c-di-GMP (Fig. 2f), this key residue for c-di-GMP binding is sterically far away from the DNA-binding motif[9]. Indeed, the K107A mutation of H-NS did not affect its binding affinities for the promoter sequences of *clpV*, *vgrG*, *fimA* and *yaiU* in the absence of c-di-GMP (Figs. 3e, 4a). In agreement, EMSA assays also demonstrated specific binding of H-NS$_{K107A}$ to the four promoters (Supplementary Fig. 15a–d). However, in contrast to

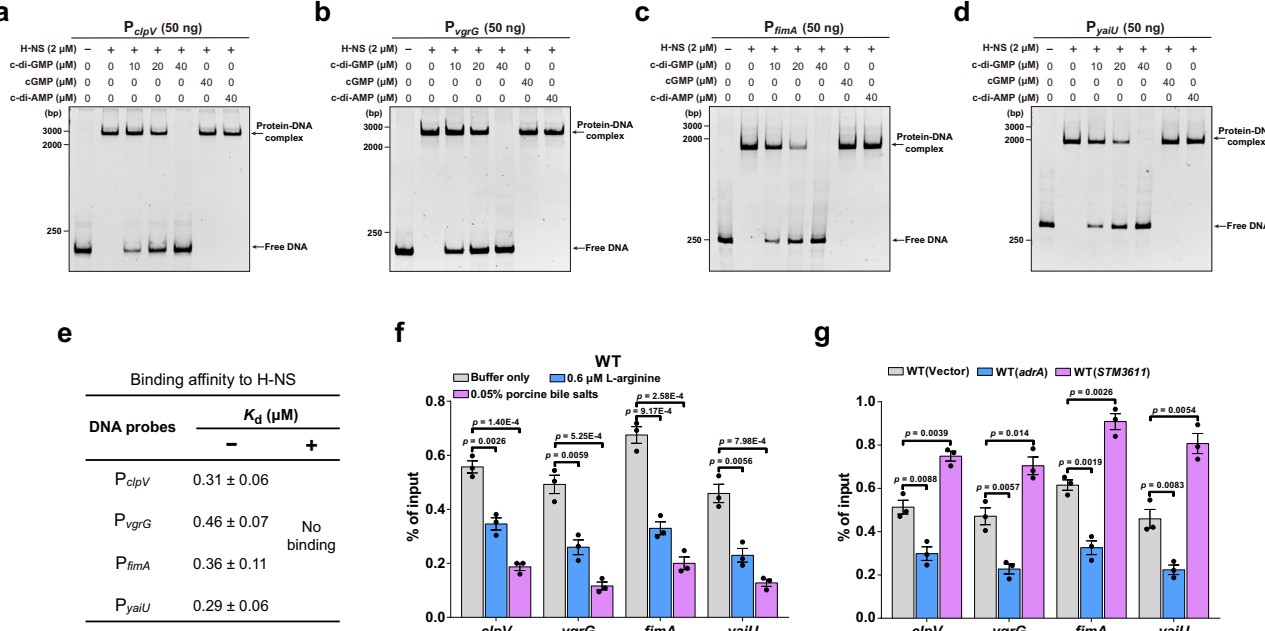

**Fig. 3 | c-di-GMP interferes with H-NS binding to DNA in *Salmonella*. a–d** EMSAs for H-NS binding to promoters of *clpV* (**a**), *vgrG* (**b**), *fimA* (**c**) or *yaiU* (**d**) in the absence or presence of nucleotides. Nucleotides were added simultaneously with H-NS to the reaction system. Gels shown are representative of three independent experiments with similar results. **e** Binding of H-NS for its target gene promoters in the absence (−) or presence (+) of c-di-GMP. The binding affinity was measured by ITC, and the $K_d$ values were presented as mean ± SD of three independent experiments. **f, g** ChIP-qPCR quantifying binding of FLAG-H-NS at the promoters of the indicated genes in wild-type (WT) *S.* Typhimurium stimulated by L-arginine or bile salts (**f**) or its derivatives overexpressing *adrA* or *STM3611* (**g**). The ChIP-qPCR signals were normalized to their respective DNA inputs. Data are mean ± SD of three biological replicates. Statistical significance was evaluated using a two-tailed unpaired Student's *t*-test and $p < 0.05$ indicate significant differences. Source data are provided as a Source Data file.

wild-type H-NS, the binding affinities of the K107A variant for the four promoters were not obviously altered upon the addition of c-di-GMP (Figs. 3e, 4a). Moreover, c-di-GMP was unable to impair the formation of H-NS$_{K107A}$-DNA complexes (Fig. 4b–e). These results suggest that changing K107 to alanine specifically impairs the ability of H-NS to bind and respond to c-di-GMP but leaves its DNA binding activity unaffected.

Then we constructed the point mutant *hns*(K107A) in the wild-type background. While the *hns*(K107A) mutant showed a similar growth rate as the wild-type strain (Supplementary Fig. 5), the expression levels of *clpV*, *vgrG*, *fimA* and *yaiU* were significantly reduced in the *hns*(K107A) mutant compared to the wild type (Fig. 4f). Meanwhile, we confirmed that H-NS$_{K107A}$ was produced at a level similar to wild-type H-NS (Supplementary Fig. 16), eliminating the possibility that the K107A substitution in H-NS enhances the abundance of the protein in cells. Furthermore, ChIP-qPCR results showed that the K107A mutation of H-NS significantly increased its occupancy at the promoter regions of the four genes (Fig. 4g). Furthermore, the binding of H-NS$_{K107A}$ to the target promoters was not affected by the addition of L-arginine or bile salts, as well as by overexpression of *adrA* or *STM3611* (Fig. 4h, i), while the expression levels of the four genes in the point mutant were not altered by the stimulation or the overexpression (Fig. 4j, k). These in vivo observations indicate that the K107A mutation of H-NS abrogates binding to c-di-GMP at its physiological concentration and leads to constitutive activity of H-NS$_{K107A}$ even at high levels of c-di-GMP.

**Elevated c-di-GMP levels globally upregulate expression of H-NS-repressed genes in *S.* Typhimurium**

Since H-NS is a global regulator that exerts transcriptional control over a large number of genes[5–7,10], we then performed RNA sequencing (RNA-seq) analysis to investigate whether changes in intracellular c-di-GMP concentration influence the expression of genes regulated by H-NS throughout the entire genome. RNA-seq results showed that 92 genes were significantly downregulated, while 333 genes were significantly upregulated (absolute log$_2$ fold change >1, Benjamini–Hochberg adjusted $p < 0.05$) in the Δ*hns* mutant compared with the wild type (Supplementary Data 1). When intracellular c-di-GMP levels in the wild-type strain were significantly increased by overexpression of *adrA*, 426 and 233 genes were significantly upregulated and downregulated, respectively (Supplementary Data 1). In particular, 245 genes repressed by H-NS were upregulated following the overexpression of *adrA* in the wild type, including genes within SPI-6 and other SPIs such as SPI-2, SPI-3 and SPI-5 (Fig. 5a, b). To validate the RNA-seq results, the expression of genes within the SPIs was measued by qRT-PCR analysis. Consistently, the H-NS-repressed genes within SPI-2, SPI-3 and SPI-5 are counter silenced by overexpression of *adrA* in the wild type (Fig. 5c). Moreover, the expression of the H-NS-repressed genes within the SPIs in the wild type was significantly increased after stimulation with L-arginine or bile salts (Fig. 5d). EMSAs also revealed that H-NS specifically binds to the promoter regions of several target genes or operons within the SPIs, and the binding was impaired by c-di-GMP in a dose-dependent fashion (Supplementary Fig. 17). Furthermore, ChIP-qPCR assays showed that overexpression of *adrA* in the wild type resulted in significantly reduced occupancy of FLAG-H-NS at the promoter regions of the target genes or operons, while K107A mutation of FLAG-H-NS significantly increased its occupancy at these promoter regions (Fig. 5e). Collectively, these results suggest that elevated c-di-GMP levels upregulate expression of H-NS-repressed genes throughout the genome in *S.* Typhimurium.

## Discussion

Transcriptional silencing of genes by H-NS was previously reported to be counteracted by environmental stimuli such as anaerobiosis, pH, temperature and osmolarity[5,6,32], but the mechanisms by which H-NS translates changes in environmental signals into altered DNA binding

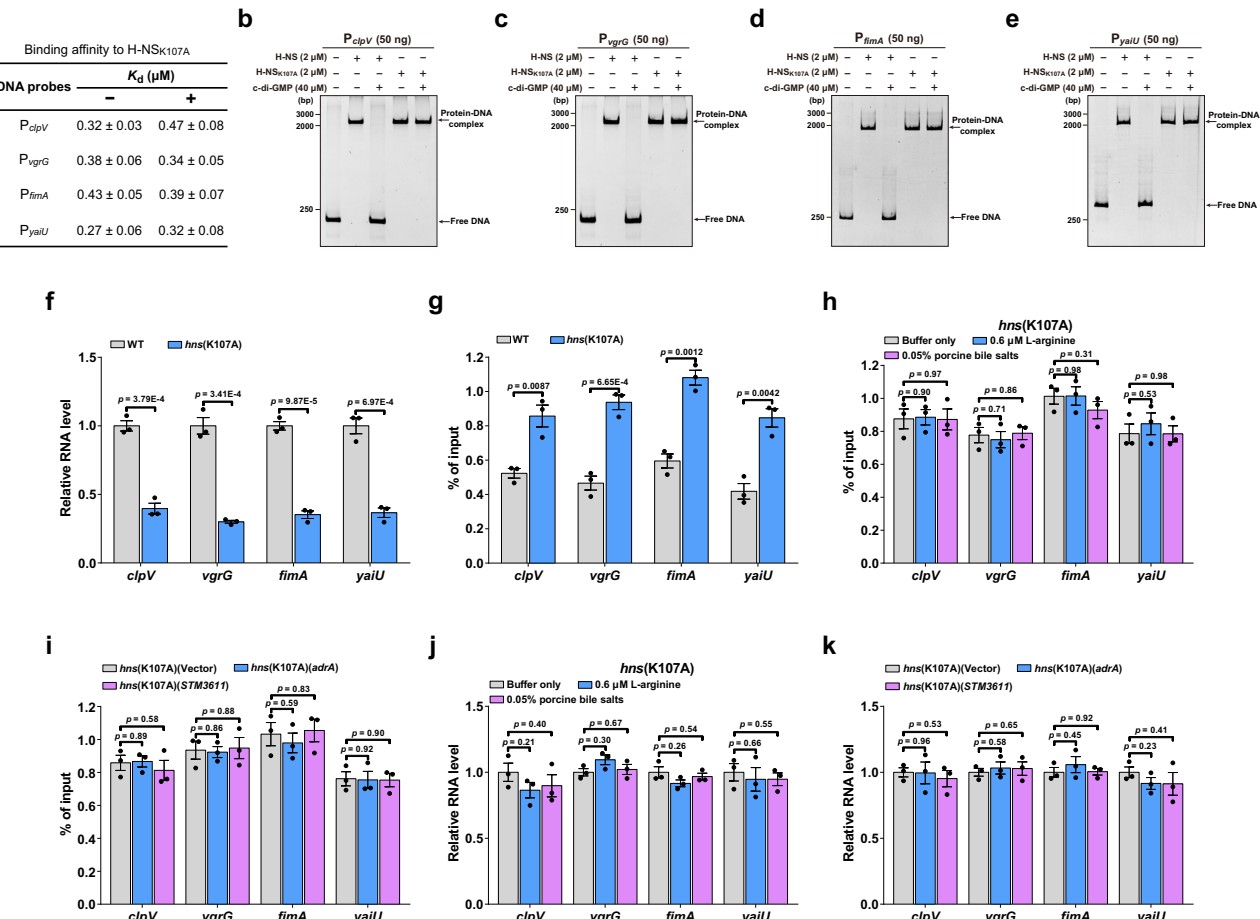

**Fig. 4 | The K107A variant remains similar DNA binding activity as wild-type H-NS but loses response to c-di-GMP in vivo. a** Binding of H-NS$_{K107A}$ for its target gene promoters in the absence (−) or presence (+) of c-di-GMP. The binding affinity was evaluated using ITC analysis, with $K_d$ values presented as mean ± SD of three independent experiments. **b**–**e** EMSAs for binding of H-NS or H-NS$_{K107A}$ to the promoters of *clpV* (**b**), *vgrG* (**c**), *fimA* (**d**) or *yaiU* (**e**) in the absence or presence of c-di-GMP. c-di-GMP was added simultaneously with the proteins to the reaction system. Gels shown are representative of three independent experiments with similar results. **f** qRT-PCR analysis of the mRNA levels of four H-NS-repressed genes in wild-type (WT) *S.* Typhimurium and the *hns*(K107A) mutant. **g** ChIP-qPCR quantifying binding of FLAG-H-NS in the WT strain and FLAG-H-NS$_{K107A}$ in the *hns*(K107A) mutant at the indicated promoters. **h**, **i** ChIP-qPCR quantifying binding of FLAG-H-NS$_{K107A}$ at its target promoters in the *hns*(K107A) mutant stimulated by L-arginine or bile salts (**h**) or its derivatives overexpressing *adrA* or *STM3611* (**i**). **j**, **k** qRT-PCR analysis of the expression of the indicated genes in the *hns*(K107A) mutant induced by L-arginine or bile salts (**j**) or its derivatives overexpressing *adrA* or *STM3611* (**k**). **f**, **j**, **k** Expression levels were presented as values relative to that of the WT (**f**), or that of *hns*(K107A) without stimulation (**j**) or gene overexpression (**k**). **g**–**i** The ChIP-qPCR signals were normalized by the input DNA. **f**–**k** Data are mean ± SD of three biological replicates. Statistical significance was calculated using the two-tailed unpaired Student's *t*-test, and differences were considered statistically significant at *p* < 0.05. Source data are provided as a Source Data file.

properties remain unclear and sometimes controversial[5,33–37]. A recent study showed that an increase in ambient temperature to 37 °C promotes the unfolding of the central dimerization domain of H-NS, thus breaking up its multimers and enabling an autoinhibitory compact H-NS conformation that blocks DNA binding[38]. A more recent study revealed the structural basis for osmotic regulation of the DNA binding properties of H-NS, with high salt-induced conformational changes of H-NS driving the switch between its DNA stiffening and bridging activities[39]. However, these environment-sensing molecular mechanisms seem incompatible with each other. As H-NS is a pleiotropic regulator that releases DNA to enable gene expression in response to various environmental factors[32], there may exist a widely conserved molecular basis for environment-sensing by H-NS. In the present study, our findings demonstrate that H-NS is a c-di-GMP effector, thus suggesting a general mechanism through which environmental stimuli derepress H-NS-mediated transcriptional silencing.

In addition to protein-independent mechanism that causes changes in local DNA structure to derepress H-NS-silenced genes in response to environmental signals such as temperature and osmolarity[5,6,38,39], several protein-dependent mechanisms counteracting H-NS-dependent silencing have also been reported[12]. DNA-binding anti-silencing proteins such as RovA, VirB, LeuO and SlyA are able to displace H-NS from DNA, whereas the Lon protease was shown to degrade displaced H-NS[12,40]. Moreover, formation of heterodimers between H-NS and its partial paralogues such as Hha, YdgT, YmoA, or H-NST interferes with or enhances the ability of H-NS to repress transcription[11,12,41]. Our current study shows that c-di-GMP binds free H-NS and interferes with its binding to DNA, thus revealing a previously unrecognized protein-independent derepression mechanism. In contrast to the anti-silencing proteins, c-di-GMP has no ability to displace H-NS from DNA. As the Lon protease can not degrade DNA-bound H-NS[40], c-di-GMP and Lon act to upregulate H-NS-silenced genes by two independent modes of H-NS antagonism while the protein is off the DNA. Our results also identify an H-NS variant K107A that remains similar DNA binding activity as wild-type H-NS but has lost the ability to bind and respond to c-di-GMP in cells. Nevertheless, it is expectable that this H-NS variant can still be displaced from DNA by anti-silencing proteins.

The StpA protein is a full-length paralogue of H-NS in *Salmonella* and they can form heterodimers[29,42]. In contrast to H-NS, the DNA

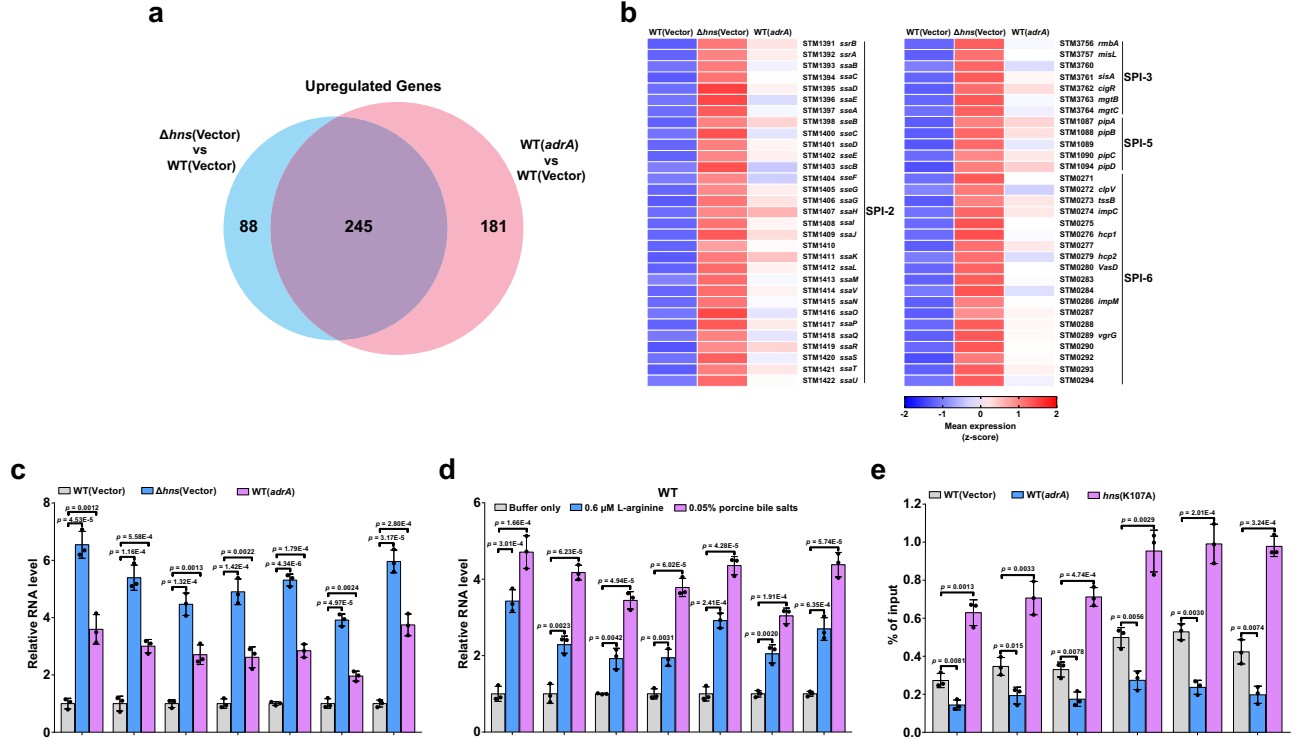

**Fig. 5 | Elevated c-di-GMP levels upregulate expression of H-NS-repressed genes throughout the genome in *S.* Typhimurium. a** Venn diagram showing the number of unique and shared genes upregulated ($\log_2$ fold change >1, Benjamini–Hochberg adjusted $p < 0.05$) in $\Delta hns$ or the wild type (WT) overexpressing *adrA* compared to the WT based on RNA-Seq ($n = 3$ biological replicates). *P* values were calculated using two-sided Wald test by the nbinomWaldTest function in DESeq2 R package and then adjusted for multiple testing using the Benjamini–Hochberg correction method. **b** Heatmap showing z-scores of normalized, $\log_2$-transformed and scaled expression of genes within the SPIs upregulated in $\Delta hns$ or the WT overexpressing *adrA* compared to the WT in RNA-Seq data ($n = 3$ biological replicates). Red represents higher abundance, and blue represents lower abundance. **c, d** qRT-PCR analysis of the expression of genes within SPI-2, SPI-3 and SPI-5 in the WT, $\Delta hns$ and the WT overexpressing *adrA* (**c**), and in the WT stimulated by L-arginine or bile salts (**d**). Expression levels were presented as values relative to that of the WT without gene overexpression (**c**) or stimulation (**d**). **e** ChIP-qPCR quantifying binding of FLAG-H-NS in the WT and its derivative overexpressing *adrA* at the promoter regions of the target genes or operons within SPI-2, SPI-3 and SPI-5, as well as binding of FLAG-H-NS$_{K107A}$ in the *hns*(K107A) mutant at these promoter regions. ChIP-qPCR signals were calculated as percentage of input. **c**–**e** Data are mean ± SD of three biological replicates and $p < 0.05$ was considered statistically significant (two-tailed unpaired Student's *t*-test). Source data are provided as a Source Data file.

binding activity of StpA should not be impaired by c-di-GMP due to the low binding affinity between them. The interference with H-NS binding to DNA by c-di-GMP may result in a failure of the H-NS-StpA heterodimers to participate in DNA-protein-DNA bridging, thus altering the structure of the heteromeric protein-DNA complexes and affecting expression of genes controlled by the heterodimers. However, while having no influence on H-NS abundance (Supplementary Fig. 7), changes in intracellular c-di-GMP concentration are believed not to affect the protective effect of H-NS against Lon-mediated degradation of StpA[40].

In conclusion, we identify H-NS as a c-di-GMP effector and provide mechanistic insights into how H-NS-mediated repression of transcription might be counteracted. While cellular levels of c-di-GMP are regulated in response to various environmental cues, our work paves the way for further exploration of environmental conditions that alleviate the H-NS-mediated silencing of genes involved in important biological processes in bacteria via c-di-GMP signaling.

## Methods
### Bacterial strains, plasmid constructions and growth conditions
Bacterial strains and plasmids used in this study are listed in Supplementary Table 1. All primers used in this study were designed using Primer premier 5.0 (Premier Biosoft) and their sequences are listed in Supplementary Table 2. *S.* Typhimurium strain SL1344, *E. coli* strains and their derivatives were grown in LB medium with appropriate

antibiotics at 37 °C, unless specified otherwise. Cellular growth was monitored based on the optical density (OD) at 600 nm. When required, antibiotics were added at the following concentrations: streptomycin, 20 µg ml$^{-1}$; kanamycin, 50 µg ml$^{-1}$; chloramphenicol, 20 µg ml$^{-1}$. For protein secretion experiments, the pKT100 derivative containing C-terminal VSVG-tagged Hcp1 was constructed. In-frame deletion and point mutants of *S.* Typhimurium were constructed using the CRISPR-Cas9 system[43]. For complementation or overexpression, derivatives of pBBR1MCS1 were transformed into relevant strains, and the expression was induced by addition of 0.5 mM isopropyl β-D-1-thiogalactopyranoside (IPTG) at the time of inoculation. For *S.* Typhimurium strains stimulated with L-arginine or porcine bile salts (Sigma-Aldrich), substances were added at OD$_{600}$ of 1.2. Unless otherwise indicated, cultures of *S.* Typhimurium strains were harvested at OD$_{600}$ of 2.5.

### Western blot analysis
After incubation, 1 ml of culture was centrifuged, and the cell pellet was resuspended in 80 µl SDS-loading buffer as the intracellular sample. 200 ml of the culture was centrifuged for 30 min at $8000 \times g$, and then the supernatant was centrifuged for another 40 min at $10,000 \times g$. The resulting supernatant was filtered through a 0.22 µm filter (Millipore), followed by three times repetitive filtration through a nitrocellulose filter (BA85) (Whatman)[44]. The filter was soaked in 80 µl of SDS-loading buffer for 15 min at 70 °C and then boiled for 15 min to resuspend

secretion proteins. For western blots, proteins in samples were resolved by SDS-PAGE and transferred onto PVDF membranes (Millipore). The membrane was blocked in QuickBlock™ Blocking Buffer (Shanghai Beyotime Biotechnology, China) for 6 h at 4 °C, and incubated overnight at 4 °C with the corresponding primary antibody: mouse anti-VSVG (Abways, Shanghai, China, cat# AB0063), 1:5,000; rabbit anti-ICDH[45], 1:5000; mouse anti-FLAG, 1:5,000 (Abways, Shanghai, China, cat# AB0028). Then, the membrane was washed five times in TBST buffer (50 mM Tris, 150 mM NaCl, 0.05% Tween 20, pH 7.4), and incubated with 1:10,000 dilution of goat anti-mouse horseradish peroxidase-conjugated secondary antibodies (DIYIBIO, China, cat# DY60203) or goat anti-rabbit horseradish peroxidase-conjugated secondary antibodies (DIYIBIO, China, cat# DY60202) for 6 h at 4 °C. After washing seven times with TBST buffer, signals were detected using the ECL kit (GE Healthcare) following the manufacturer's protocol. When required, the band intensities were quantified by scanning densitometry using ImageJ v1.52a (NIH, USA) and normalized to intracellular ICDH. Uncropped images of blots can be found in the Source Data file.

### Extraction and quantification of intracellular c-di-GMP levels

*S*. Typhimurium strains were cultured at 37 °C in LB broth with shaking at 200 rpm. 15 ml of each culture was centrifuged and the cell pellets were washed twice with phosphate-buffered saline (PBS, pH 7.0), followed by resuspension in 500 μl of extraction solution (acetonitrile/methanol/water, 40:40:20, v/v/v) and cooling on ice for 15 min. The samples were boiled for 10 min at 95 °C and then cooled again on ice for another 15 min. After centrifugation at $20,000 \times g$ and 4 °C for 10 min, supernatants were transferred to new tubes on ice. The remaining pellets were used for two more extraction steps with 500 μl of the extraction solution without heating. The pooled supernatants for each sample were lyophilized, followed by resuspension in 200 μl of distilled water. The c-di-GMP concentrations were measured by LC-MS/MS[46]. Intracellular levels of c-di-GMP were then normalized to the number of bacterial cells for each sample.

### qRT-PCR analysis

Cultures of *S*. Typhimurium strains were harvested, and then total RNA was extracted by RNAprep Pure Cell/Bacteria Kit (Tiangen Biotech, Beijing, China). After treatment with RNase-free DNase I (Sigma-Aldrich), cDNA was reverse transcribed from RNA by TransScript II One-Step gDNA Removal and cDNA Synthesis SuperMix (TransGen Biotech, Beijing, China). qRT-PCR was performed and the relative abundance of 16 S rRNA was used as an internal standard.

### Construction of promoter-*lacZ* fusion reporter strains and β-galactosidase assays

The promoter regions of *clpV*, *hcp1*, tae4, and *vgrG* were amplified by PCR and separately cloned into the *lacZ* fusion reporter vector pDM4-*lacZ*[4]. Recombinant plasmids were transformed into *E. coli* strain S17-1λpir and then transferred to *S*. Typhimurium strains by conjugation. The transconjugants were selected on LB agar containing streptomycin and chloramphenicol. The *lacZ* fusion reporter strains were grown in LB broth at 37 °C until $OD_{600}$ reached 2.5, and then the β-galactosidase activity was assayed using ONPG (o-nitrophenyl-β-d-galactopyranoside) as the substrate according to the Miller method[47].

### Overexpression and purification of recombinant proteins

To express and purify $His_6$-tagged recombinant proteins, the pET-28a derivatives were transformed into *E. coli* strain BL21(DE3). Bacteria were cultured at 37 °C in LB medium to an $OD_{600}$ of 0.5, and then 0.25 mM IPTG was added to induce protein expression at 22 °C for an additional 12 h. Harvested cells were lysed by sonication in the lysis buffer (20 mM Tris, 150 mM NaCl, pH 7.5) and then unbroken cells and debris were removed by centrifugation. Recombinant proteins in the

supernatant were then purified with the His-Bind Ni-NTA resin (Novagen, Madison, WI) according to the manufacturer's instructions, followed by dialysis against a Tris buffer (50 mM Tris, 150 mM NaCl, 10% glycerol, pH 7.5) at 4 °C. Protein purity was evaluated by SDS-PAGE and protein concentration was determined using the Bradford assay with bovine serum albumin (BSA) as standard. After being concentrated using an Amicon Ultra-4 10 kDa cutoff centrifugal filter (Millipore), the purified H-NS and its variants were further subjected to SEC using an ÄKTA Purifier FPLC system (GE Healthcare). A Superdex 200 16/600 GL gel filtration column was equilibrated with the SEC buffer (20 mM Tris-HCl, 250 mM NaCl, 1 mM DTT, 10% glycerol) at a flow rate of 1 ml min⁻¹ at room temperature. When needed, the N-terminal $His_6$ tag of the purified proteins was cleaved by thrombin (Sigma-Aldrich), and then the label-free proteins were obtained by the second round of Ni-NTA affinity chromatography followed by the removal of thrombin with benzamidine sepharose (GE Healthcare).

### UV-crosslinking assays

20 μM $His_6$-tagged proteins were incubated with 5 μM biotinylated c-di-GMP (Biolog, Germany) in a Tris buffer (50 mM Tris-HCl, 50 mM NaCl, 5 mM $MgCl_2$, pH 7.5) for 15 min on ice, and then subjected to UV radiation for 30 min[48]. When needed, unlabeled c-di-GMP, cGMP or c-di-AMP were added to the reaction mixture together with biotinylated c-di-GMP. After UV exposure, reaction samples were resolved by SDS-PAGE and transferred onto nitrocellulose membranes (Millipore). The membrane was then subjected to UV irradiation for another 10 min in 0.5×TBE buffer (44.5 mM Tris-HCl, 44.5 mM NaCl, 10 mM EDTA, pH 8.0), followed by blocking in the Blocking Buffer (Thermo Scientific) for 4 h at 4 °C. Subsequently, the membrane was incubated with 1:10,000 dilution of streptavidin-horseradish peroxidase (Thermo Scientific, cat# 21126) for 6 h at 4 °C. After washing four times with the Wash Buffer (Thermo Scientific), chemiluminescent signals were detected using the ECL kit (GE Healthcare). For control, a membrane exposed to UV irradiation was blocked, incubated first with 1:5,000 dilution of mouse anti-His (Abways, Shanghai, China, cat# AB0002), followed by incubation with 1:10,000 dilution of goat anti-mouse horseradish peroxidase-conjugated secondary antibodies (DIYIBIO, China, cat# DY60203). The signals were also detected using the ECL system. Uncropped images of blots can be found in the Source Data file.

### ITC analysis

ITC measurements were performed at 20 °C using a Nano ITC Standard Volume isothermal calorimeter (TA Instruments, New Castle, DE). For titrations with nucleotides, the tag-free proteins were dialyzed with a Tris buffer (25 mM Tris-HCl, 150 mM NaCl, pH 7.5) and diluted to 10 μM. 100 μM c-di-GMP, cGMP and c-di-AMP (Sigma-Aldrich) used for titrations were diluted with the same buffer. For titrations with DNA probes, promoter sequences were amplified by PCR and then purified on 6% native polyacrylamide gels. The DNA probes and proteins used for titrations were diluted with the Tris buffer to 100 μM and 10 μM, respectively. For competitive binding, c-di-GMP at a final concentration of 50 μM was added to both DNA probes and proteins involved in the titrations. All samples were degassed prior to titrations. Control experiments were also performed, with the ligand solution titrated into the buffer in the sample cell. After subtracting the heat of dilution from the experimental titrations, microcalorimetric data were fit to an independent binding model by the NanoAnalyze software version 3.4 (TA Instruments)[49].

### Molecular docking analysis

The solution structure of the C-terminal domain of *S*. Typhimurium H-NS was retrieved from the Protein Data Bank under the accession code 2L93 (https://www.rcsb.org/structure/2L93). Potential pockets and cavities were predicted using the web-based POCASA 1.1 with a

probe radius of 2 Å[26]. The 3D structure of c-di-GMP was extracted from the crystal structure of the c-di-GMP-MapZ complex (PDB ID: 2L74; https://www.rcsb.org/structure/2L74) and its flexible torsions were assigned using Autodock4[50]. Docking simulation was done by using AutoDock Vina 1.1.2[27], with the best binding mode selected based on the lowest docking energy. The three-dimensional figure was displayed with PyMOL v2.5.2 (http://www.pymol.org) and the protein-ligand interaction was analyzed using LigPlot+ v2.2.4[51].

## EMSAs

DNA probes for EMSAs were amplified by PCR and purified on 6% native polyacrylamide gels. In a 20-μl reaction system, 50 ng of DNA probes were mixed with tag-free proteins in a Tris buffer (20 mM Tris-HCl, 4 mM $MgCl_2$, 100 mM NaCl, 1 mM dithiothreitol, 1% NP-40 and 10% glycerol, pH 7.4). When needed, nucleotides (c-di-GMP, cGMP or c-di-AMP) were added simultaneously with H-NS to the reaction system. After incubation for 30 min at room temperature, the reaction mixtures were subjected to 6% native polyacrylamide gel and run in the 0.5×TBE buffer at 100 V. The DNA probe was detected using SYBR Safe DNA gel stain (Invitrogen) and imaged by the Tanon gel analysis software version 2.30 (Tanon 5200Multi, China). Competitive EMSAs were also performed incubating first H-NS with DNA probes for 20 min, followed by the subsequent addition of c-di-GMP and incubation for another 20 min. Uncropped images of gels can be found in the Source Data file.

## RNA-seq experiments

*S.* Typhimurium strains were inoculated into fresh LB broth with 0.5 mM IPTG, and then cultured at 37 °C with shaking at 200 rpm to an $OD_{600}$ of 2.5. Bacterial cells were harvested and then total RNA was extracted using RNeasy mini kit (Qiagen). Genomic DNA was degraded using the TURBO DNA-free™ kit (Life Technologies). Subsequently, ribosomal RNA was removed using MICROBExpress™ Bacterial mRNA Enrichment Kit (Life Technologies). cDNA libraries were prepared using Bacterial ScriptSeq Complete Kit (Illumina) following the manufacturer's instructions, and sequencing was performed by Sangon Biotech Co., Ltd. (Shanghai, China). RNA-seq was performed on triplicate samples. Sequence reads were mapped to the *S.* Typhimurium LT2 reference genome (NC_003197 [https://www.ncbi.nlm.nih.gov/nuccore/NC_003197] and NC_003277 [https://www.ncbi.nlm.nih.gov/nuccore/NC_003277]) using Bowtie (version 2.3.2). FeatureCounts v1.6.0 was used to count the read numbers mapped to each gene. Relative transcript abundance was calculated based on the number of reads per kilobase per million mapped sequence reads (RPKM). Differential expression analysis was performed using the DESeq2 R package (version 1.12.4), with a threshold set to $p < 0.05$ and $|\log_2$ fold-change$| > 1$. *P* values were adjusted using the Benjamini–Hochberg approach[52] to control the false discovery rate (FDR).

## ChIP-qPCR

*S.* Typhimurium strains expressing in situ tagged FLAG-H-NS and FLAG-H-NS$_{K107A}$ were generated by using the CRISPR-Cas9 system[43]. Bacterial cultures were treated with 1% formaldehyde for 15 min at room temperature. The crosslinking reaction was stopped by addition of 125 mM glycine. Bacterial pellets were washed twice with PBS, resuspended in lysis buffer (25 mM Tris-HCl, pH 7.5; 150 mM NaCl; 1 mM EDTA; 0.1% Triton X-100; 0.1% SDS), and then sonicated to generate DNA fragments of 100–500 bp. Cell debris was removed by centrifugation at 4 °C, and the supernatant was used as input sample for the IP experiments. Subsequently, the input sample was incubated with either no antibody (mock-IP) or mouse anti-FLAG (1:1,000 dilution, Abways, Shanghai, China, cat# AB0028), followed by enrichment with Protein A magnetic beads (Sigma-Aldrich)[53]. Immunoprecipitated

Protein-DNA complexes were eluted from the beads using elution buffer (50 mM Tris-HCl, pH 7.5; 10 mM EDTA; 1% SDS) at 65 °C for 30 min. Crosslinks were then reversed by incubation for 6 h at 65 °C in 0.5× elution buffer plus 250 μg ml$^{-1}$ proteinase K. DNA was extracted twice with phenol/chloroform (1:1), precipitated and resuspended in distilled water[54]. Quantification of H-NS- and H-NS$_{K107A}$-bound DNA was carried out by qPCR using KAPA SYBR FAST qPCR Kit (Kapa Biosystems) in a Roche LightCycler 96 Real-Time PCR system. The qPCR signals were normalized to input DNA[55]. Background signals from mock samples without the addition of the anti-FLAG antibody were subtracted in the final analysis.

## Statistical analysis

GraphPad Prism Software (GraphPad Prism 8.00) or Microsoft Excel 2019 was used to perform statistical analyses. Data from RNA-seq were analyzed using two-sided Wald test by the nbinomWaldTest function in DESeq2 R package and the resulting *p* values were adjusted using Benjamini–Hochberg FDR correction. All other experiments were analyzed using the two-tailed unpaired Student's *t*-test, and data are presented as mean ± SD. Statistical significance was defined as $p < 0.05$.

## Reporting summary

Further information on research design is available in the Nature Portfolio Reporting Summary linked to this article.

# Data availability

The protein 3D coordinate data used in this study are available in the PDB database under the accession code 2L93. The structure of c-di-GMP used in this study is available from the crystal structure of the c-di-GMP-MapZ complex in the PDB database under the accession code 2L74. The *S.* Typhimurium LT2 reference genome used in this study is available in the NCBI nucleotide database under the accession numbers NC_003197 and NC_003277. The RNA-seq data generated in this study have been deposited in the NCBI BioProject database under the accession number PRJNA975738. All the other data that support the findings of this study are available within the paper and its Supplementary Information and Supplementary Data. Source data are provided with this paper.

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

## Acknowledgements

This work was supported by grants from the National Key R&D Program of China (2018YFA0901200 to X.S. and L.Z.), the National Natural Science Foundation of China (32370134 and 32170048 to L.Z., and 41876114 to Q.Y.), the Natural Science Basic Research Program of Shaanxi (2023-JC-JQ-18 to L.Z.), and the Science Foundation of Donghai Laboratory (DH-2022KF0218 to Q.Y.). We thank the Teaching and Research Core Facility at College of Life Science (Ningjuan Fan and Hui Duan) and Life Science Research Core Services (Luqi Li), Northwest A&F University for technical support.

## Author contributions

Author contributions: L.Z. designed research; S.L., Q.L., C.D., J.L., H.S., L.X., and Q.Y. performed research; L.Z., S.L., and Y.W. analyzed data; and L.Z., X.S., and S.L. wrote the paper.

## Competing interests

The authors declare no competing interests.
