## [Peer Review File · Nature Communications]

c-di-GMP inhibits the DNA binding activity of H-NS in SalmonellaReviewer #1 (Remarks to the Author):

This is an interesting manuscript that reports a new regulatory mechanism, and thus is potentially a good fit for Nature Communications. Essentially, it argues one key point, that a secondary messenger c-di-GMP directly binds to H-NS and H-NS-like proteins in Gram-negative bacteria and relieves xenogeneic silencing by interfering with H-NS binding to its target DNAs. The authors focus on T6SS in *Salmonella*, which is important for effector delivery into bacterial and eukaryotic cells, but H-NS and other NAPs control large numbers of genes, many of which are implicated in pathogenesis and responses to environmental signaling.

The data are convincing, controls are in place, and approaches are rigorous. Classical genetic data indicate that c-di-GMP activates the T6SS expression in *Salmonella*, and that this activation is due to the relief of H-NS-mediated silencing. Based on docking, subsequently validated by mutagenesis, the authors conclude that H-NS directly binds c-di-GMP. Since this binding site partially overlaps with a well-defined DNA-binding interface, their findings that c-di-GMP interferes with H-NS-DNA interactions are not surprising, but are very well documented.

The manuscript is overall well written, and figures are easy to follow. On the flip side, the manuscript is too long, and the figures are too many, in the end distracting from the main points. There are numerous redundancies throughout, and statements that lead nowhere, only add references. For example, starting on line 59 "RsmA acts as a negative translational regulator...ref 17,18" – this is irrelevant for the current study. Along the same lines, the conclusions are presented in too many words. This is a straightforward and logical story, and the data speak for themselves – there is no need to explain every data point, figures exist for this purpose. The approaches are standard, for example published by the authors in Nat Comm 2022 paper on T3SS control, and do not require a lot of explanations. In my opinion, the authors should go through their manuscript, eliminate unnecessary statements, and simplify the necessary ones – from all sections, Abstract through Discussion.

The same applies to figures, particularly in the Supplementary Data. There are many ITC figures, and they are unnecessary – ITC is not a novel method and, once shown (as in Fig. 3bc), these results should be summarized in a table (as done in Fig. 3f). Errors should be added to the table, and perhaps stoichiometry coefficients as well.

General points that could benefit from explanation/clarification:

The authors show that L-Arg and bile salts induce c-di-GMP production, but bile salts have a more potent effect. What is the reason to use both experimentally?

The cellular concentration of c-di-GMP has been determined in *Salmonella*. The range needs to be stated.

In Fig. 5, the authors compare genes that are controlled by H-NS and c-di-GMP. It is not surprising that c-di-GMP has many additional targets, as it binds to other proteins. But what can be inferred from the identity of genes uniquely controlled by H-NS, potentially independently from its DNA-binding activity?

In competition experiments, are *E. coli* cells dying (lysing) or simply fail to grow? And what is the reason to use DH5a? This is not a good example of a commensal gut bacterium.

The authors show that *V. parahaemolyticus*, *P. aeruginosa* and *P. putida* also bind c-di-GMP, establishing this interaction as a general regulatory principle. Quite unexpectedly, however, the binding sites are not at all conserved, even if their importance is supported by experimental data. I think that more discussion of this point is warranted and perhaps analogous cases could be brought up. In general, given this diversity, I found the analysis of "other" H-NS silencers to be more distracting than helpful because it raises a long list of questions.

Reviewer #2 (Remarks to the Author):

This paper reports the important finding that c-di-GMP is a ligand that interferes with the H-NS protein's ability to bind DNA. This discovery has implications for our understanding of the very many bacterial systems where H-NS-like nucleoid-associated proteins act to silence transcription. We must now take into account the possibility that the c-di-GMP second messenger is influencing their expression via modulation of the DNA binding activity of H-NS.

As written, the key message of the paper concerning H-NS modulation by direct interaction with c-di-GMP is submerged in a detailed story about type 6 secretion system (T6SS) gene regulation in Salmonella. Because far more investigators study H-NS than study Salmonella T6SS, the impact of the authors' findings may be diminished if the report retains its current structure and title. Perhaps the authors could consider amending both to give more emphasis to the c-di-GMP and H-NS aspect? One might even be tempted to turn this report into two papers, one for a general journal such as Nature Comms on c-di-GMP binding to H-NS, and a separate paper for a microbial pathogenesis journal on T6SS regulation.

Throughout the study, the experiments have been executed with great thoroughness. Appropriate controls are included, together with statements of reproducibility and, where required, statistical analysis of the data is provided. The authors place their findings in physiological context and consider their implications for our knowledge of Salmonella virulence and of competition with other gut microbes. However, there remain some surprising gaps in the investigation and the authors should be encouraged to address these prior to publication.

One gap concerns the mechanism by which c-di-GMP affects H-NS activity. The paper contains copious data showing that the ligand interferes with H-NS binding to DNA. However, the reader will also want to know if c-di-GMP can displace H-NS from DNA. This requires experiments with H-NS prebound to DNA targets that are treated with increasing concentrations of c-di-GMP. The point is important because we currently have two broad categories of interference with H-NS binding to DNA. One interference mechanism involves displacement of H-NS from DNA, usually by another DNA binding protein (PMID: 18757787) while the other mechanism involves inhibition of H-NS DNA binding activity while the H-NS protein is DNA-free. The Lon protease operates in this second way in Salmonella (PMID: 32209674). Could the Lon protease and c-di-GMP act to upregulate H-NS-silenced genes by contributing two independent modes of H-NS inhibition while the protein is off the DNA?. Both of these mechanisms have the effect of reducing the pool of DNA-binding competent H-NS in the cell, increasing the probability that silenced genes will become transcribed. (They are distinct from mechanisms in which H-NS interaction with proteins of the Hha family influences DNA target selection by H-NS.) Some new EMSA experiments will allow the authors to test the c-di-GMP-dependent H-NS displacement hypothesis.

A second gap arises because the authors do not seem to address the presence in Salmonella of StpA, a closely-related paralogue of H-NS, that is encoded by the chromosomally-located gene, stpA. Given the wide sweep of H-NS-like proteins that the authors have considered in their analysis, it seems strange that they have not mentioned this second H-NS-like protein in Salmonella. Of the five amino acids in Salmonella H-NS implicated in c-di-GMP binding that are highlighted in yellow in Fig 8c, two are not present in StpA. The tyrosine is replaced by phenylalanine and the valine by threonine. In the following alignment, the five amino acids are bracketed by spaces on either side to facilitate identification:

```
H-NS(St) 91-AARPAKYS Y V D ENGET K TWTGQGR T-115  
StpA(St) 91-QPRPAKYR F T D FNGEE K TWTGQGR T-115
```

Lysine K107 is present in both proteins.

Given the ability of StpA to form heterodimers with H-NS, readers will be curious to know if c-di-GMP can influence the binding of StpA to DNA, either as an StpA homodimer or as a heterodimer with H-NS. The heterodimer situation becomes even more complicated in Salmonella clinical isolates that have large plasmids expressing additional H-NS-like proteins (PMID: 20444106).

There is considerable overlap between the regulons of genes controlled by StpA and H-NS and StpA has a key role in controlling the production of RpoS, a sigma factor whose gene can acquire compensatory mutations in hns knockout mutants (PMID: 19843227, PMID: 26631971). Indeed, the stpA gene is also prone to the acquisition of compensatory lesions when hns is disrupted (PMID: 25375226). Before branching out to a discussion of H-NS-like proteins in other bacterial species, perhaps the authors could make an assessment of StpA in Salmonella?

Lines 168-170: In Salmonella, the Fur protein is a negative regulator of hns transcription (PMID:

21075923, PMID: 22017966). The authors will need to adjust their description of the roles of the Fur and H-NS proteins in the control of T6SS gene expression to take into account this Fur-hns direct regulatory relationship.

Lines 179-182: "Despite the lethality of the hns null mutation in *S. Typhimurium* strain ATCC 14028s, *S. Typhimurium* strain LT2 has been shown to tolerate the hns mutation (15,26). Consistently, an hns deletion mutation was successfully constructed by the CRISPR-Cas9 system in strain SL1344." Knockout mutations in the *Salmonella* hns gene are tolerated when compensatory mutations arise either in the rpoS gene or the phoPQ operon. Strain LT2 is already defective in expressing full levels of RpoS (the stress and stationary phase sigma factor of RNA polymerase) so it tolerates inactivation of hns because, genetically, it is 'pre-compensated'. This is not the case with strain SL1344, used by the authors of the present study. Here, it is likely that the authors have isolated an SL1344 hns mutant that has compensatory lesions in either the phoPQ operon or the rpoS gene. This should be investigated because, if compensatory mutations are present, their data interpretation must be expanded to take into account the effects of these additional mutations. Both RpoS and PhoPQ exert widespread influence on the expression of *Salmonella* virulence genes, including many of the genes that the authors study in the present investigation.

It should perhaps be mentioned that the K107A mutant H-NS can still be displaced from DNA by other mechanisms (e.g. the intervention of anti-repressor proteins). These mechanisms have been described in numerous bacterial systems (PMID: 18757787).

Minor item

Line 18, Gram-negative, not 'gram-negative' (Gram is a person's name – Hans Christian Gram).

Reviewer #3 (Remarks to the Author):

This paper looks at the connection between the signaling molecule c-di-GMP and the DNA-binding, silencing factor H-NS in *Salmonella*. The authors argue that (i) c-di-GMP directly binds H-NS, preventing H-NS from binding DNA, (ii) high c-di-GMP levels lead to de-repression of H-NS-silenced genes, (iii) de-repression of H-NS-silenced genes provides a fitness advantage because of increased Type VI secretion, and (iv) the functionally analogous MvaT protein from *Pseudomonas* is also bound and inhibited by c-di-GMP. These conclusions have major implications for our understanding of the role of c-di-GMP, and for how H-NS activity is modulated. The work is generally very thorough, and I have only one suggested experiment that would add an important control:

- The authors should use ChIP-qPCR to test whether K107A H-NS binding is affected either by bile salts (or by a more direct modulation of c-di-GMP levels).

Additional comments:

1. I recommend moving Supplementary Figure 1 into Figure 1.
2. Change "Relative gene expression" labels on rtPCR graphs to "Relative RNA level".
3. Data from Supplementary Figures 11 and 13 showing that the K107A H-NS mutant is unaffected by c-di-GMP are key to the main conclusions of the paper. I recommend moving these to the main figures.
4. Many strains of LT2 are avirulent. Can the authors comment on the genotype of the strain they used?
5. Are there RNA-seq data from *Salmonella* or *Escherichia coli* with altered c-di-GMP levels? If so, the authors could use those data as an independent test of their model, since H-NS-repressed genes are well characterized in these species.
6. Can the authors comment on why increased c-di-GMP levels in the two tested *Pseudomonas* species has the same phenotype as deleting *mvaT*, despite the fact that MvaU would be expected to contribute to repression? Related to this, the authors should at least mention MvaU.
7. Deletion of hns is lethal in other *Salmonella* strains and some other species, and leads to a growth defect in *E. coli*. Can the authors speculate on the impact of raised c-di-GMP levels on

bacterial fitness, since their model indicates that high c-di-GMP levels would derepress all H-NS-silenced genes?

Reviewer #4 (Remarks to the Author):

c-di-GMP is a widespread secondary messenger in bacteria, which regulates numerous different processes, e.g. related to the planktonic/biofilm life style, motility, pathogenicity and more. Over the last decades a number of different effector proteins have been identified that bind and thereby respond to c-di-GMP. In this study, the authors convincingly demonstrate that the highly conserved histone-like nucleoid structuring protein H-NS is, in fact, a c-di-GMP-binding regulator in *Salmonella Enterica*. The authors present in vitro and in vivo approaches that demonstrate direct c-di-GMP binding to H-NS and provide evidence that the c-di-GMP-binding site partly overlaps with the DNA-binding site of H-NS, so that DNA interaction of the protein is abrogated upon rising levels of c-di-GMP. In *S. Typhimurium* they show by RNAseq approaches that a number of factors are regulated by this mechanism, among them activation of a type VI secretion system (T6SS), which occurs upon exposure of the cells to signals such as bile acids. By mouse models they show that, in fact, the signaling cascade from bile acids to T6SS activation likely functions via c-di-GMP and H-NS and provides an advantage in successful colonization of the host gut. They also present evidence that H-NS in other gammaproteobacteria such as *Vibrio parahaemolyticus*, *Pseudomonas aeruginosa* and *P. putida* binds and responds to c-di-GMP binding, so that this mechanism is likely highly conserved.

The major and highly relevant finding that highly conserved H-NS serves as a c-di-GMP-binding regulator is convincing as it is supported by the results presented by the authors. The number of high-quality experiments is impressive and the manuscript is very well structured. I have only some minor comments with respect to this study.

58, 'RsmA acts as ...'; this passage does not really fit there and should be moved to somewhere else or could be omitted completely.

153 and throughout manuscript, 'significantly' could also be true to rather small differences. I suggest to at least sometimes mention the factor of changes (e.g. double, six-fold, ..) where appropriate.

173, Fig. 2a and throughout manuscript: There are many bar diagrams that provide complex sets of data. To make it a little bit easier for the reader, the authors may consider to add more information to the figure panels, e.g., already indicate in the panel which strain is used.

233, please state that alanine substitutions were introduced.

234, and general, it would be nice to see a presentation of the protein purifications to visualize the quality of the purification. It would also be nice if, for example, a SEC elution profile for the substitution variants could be presented to demonstrate that the structure of the mutated protein is preserved.

259, is FLAG-H-NS functional?

355, I would ask the authors to add a little bit of explanation here – who is competing and under which conditions.

389, strongly, better indicate a factor here (see above)

390, Fig. S18 shows growth state-dependent, that should be mentioned here.

456, please name the conserved residue

Just as a final small remark: in basically all figures, axis labeling or the insets are sometimes really

small and hard to make out in the PDF for reviewing.

Response to Reviewers

We wish to begin by thanking the four reviewers for their very supportive and constructive comments. In the revised manuscript, we have addressed all concerns from the reviewers to our best and hope our revision and explanations could answer the reviewers' questions. Please find our detailed responses to each of the comments below.

Reviewer #1 (Remarks to the Author):

This is an interesting manuscript that reports a new regulatory mechanism, and thus is potentially a good fit for Nature Communications. Essentially, it argues one key point, that a secondary messenger c-di-GMP directly binds to H-NS and H-NS-like proteins in Gram-negative bacteria and relieves xenogeneic silencing by interfering with H-NS binding to its target DNAs. The authors focus on T6SS in *Salmonella*, which is important for effector delivery into bacterial and eukaryotic cells, but H-NS and other NAPs control large numbers of genes, many of which are implicated in pathogenesis and responses to environmental signaling.

Response: The authors are very grateful for the positive feedback provided by the reviewer, as well as the insightful comments and constructive suggestions. As suggested, we have refocused the paper on the main finding that c-di-GMP modulates DNA binding activity of H-NS in *Salmonella*.

The data are convincing, controls are in place, and approaches are rigorous. Classical genetic data indicate that c-di-GMP activates the T6SS expression in *Salmonella*, and that this activation is due to the relief of H-NS-mediated silencing. Based on docking, subsequently validated by mutagenesis, the authors conclude that H-NS directly binds c-di-GMP. Since this binding site partially overlaps with a well-defined DNA-binding interface, their findings that c-di-GMP interferes with H-NS-DNA interactions are not surprising, but are very well documented.

Response: We thank the reviewer for the supportive and constructive comments.

The manuscript is overall well written, and figures are easy to follow. On the flip side, the manuscript is too long, and the figures are too many, in the end distracting from the main points. There are numerous redundancies throughout, and statements that lead nowhere, only add references. For example, starting on line 59 “RsmA acts as a negative translational regulator...ref 17,18” – this is irrelevant for the current study. Along the same lines, the conclusions are presented in too many words. This is a straightforward and logical story, and the data speak for themselves – there is no need to explain every data point, figures exist for this purpose. The approaches are standard, for example published by the authors in Nat Comm 2022 paper on T3SS control, and do not require a lot of explanations. In my opinion, the authors should go through their manuscript, eliminate unnecessary statements, and simplify the necessary ones – from all sections, Abstract through Discussion.

Response: We thank the reviewer for the insightful comments and constructive suggestions. We have eliminated unnecessary statements and simplified the necessary ones from all sections in the revised manuscript.

The same applies to figures, particularly in the Supplementary Data. There are many ITC figures, and they are unnecessary – ITC is not a novel method and, once shown (as in Fig. 3bc), these results should be summarized in a table (as done in Fig. 3f). Errors should be added to the table, and perhaps stoichiometry coefficients as well.

Response: We thank the reviewer for this important point. As suggested, we have summarized ITC results in a table in **Figs. 2f, 3e, 4a** and **Supplementary Figs. 10b, 13**.

General points that could benefit from explanation/clarification:

The authors show that L-Arg and bile salts induce c-di-GMP production, but bile salts have a more potent effect. What is the reason to use both experimentally?

Response: We thank the reviewer for raising this important question. As both L-arginine and bile salts have been reported to induce c-di-GMP production in *Salmonella* in previous studies (Li et al., Nat Commun. 2022, 13:6684; Mills et al., Sci Signal. 2015, 8:ra57), we used these two substrates to demonstrate that elevated intracellular c-di-GMP levels induced by different environmental signals can upregulate expression of H-NS-regulated genes. The more potent effect of bile salts than L-arginine may be attributed to their different concentrations used as well as their different mechanisms to induce c-di-GMP production in *Salmonella*. Bile components taurocholate and taurodeoxycholate were shown to increase c-di-GMP concentrations via binding to the sensory domain of the DGC YedQ (Li et al., Nat Commun. 2022, 13:6684), while L-arginine was reported to stimulate the DGC activity of YedQ in an indirect way (Mills et al., Sci Signal. 2015, 8:ra57).

The cellular concentration of c-di-GMP has been determined in *Salmonella*. The range needs to be stated.

Response: We thank the reviewer for this important point. As it is difficult to determine cell volumes, absolute quantification of the cellular level of c-di-GMP as volumetric molar concentration was performed by a FRET-based c-di-GMP biosensor, which characterizes the amount of c-di-GMP in live bacteria using microscopy or flow cytometry (Mills et al., Sci Signal. 2015, 8:ra57; Petersen et al., Proc. Natl. Acad. Sci. U S A. 2019, 116:6335-6340). However, the method is complicated and difficult to master, while the amounts of c-di-GMP in individual cells vary greatly. In the present study, we adopted a widely used relative quantitative method and intracellular levels of c-di-GMP are normalized to the number of bacterial cells (Almblad et al., Nat Commun. 2021, 12:1986; Li et al., Nat Commun. 2022, 13:6684). As suggested, the

approximate concentration range of c-di-GMP in *Salmonella* as determined by the FRET-based c-di-GMP biosensor have been given (**Lines 193-194**).

In Fig. 5, the authors compare genes that are controlled by H-NS and c-di-GMP. It is not surprising that c-di-GMP has many additional targets, as it binds to other proteins. But what can be inferred from the identity of genes uniquely controlled by H-NS, potentially independently from its DNA-binding activity?

Response: We thank the reviewer for this very insightful point. Indeed, in addition to functioning as a repressor by binding DNA directly, H-NS was also shown to mediate gene expression indirectly via regulating expression of transcription factors (Navarre et al., *Science*. 2006, 313:236-238) or interacting with its full-length or partial paralogues such as StpA, Sfh, Hha, YdgT, YmoA or H-NST (Stoebel et al., *Microbiology*, 2008, 154:2533-2545). For instance, the H-NS oligomerization domain alone can silence expression of some genes via interacting with its full-length paralogue StpA in *E. coli* (Free & Dorman, *J Bacteriol.* 1997, 179:909-918; Free et al., *Mol Microbiol.* 2001, 42:903-917). On the other hand, while deletion of *hns* completely abolished the repressive effect of H-NS on its target genes, elevated intracellular levels of c-di-GMP only partially relieved the repressive effect of H-NS. When the statistical significance threshold was set to $p < 0.05$ and $|\log_2 \text{fold-change}| > 1$, some genes upregulated more than 2-fold in Δhns VS WT, but upregulated less than 2-fold in WT VS WT(*adrA*) were identified as genes uniquely controlled by H-NS in the present study.

In competition experiments, are *E. coli* cells dying (lysing) or simply fail to grow? And what is the reason to use DH5a? This is not a good example of a commensal gut bacterium.

Response: We thank the reviewer for raising this important question. Whether *E. coli* cells are dying (lysing) or fail to grow is dependent on the function of

T6SS effector proteins that are delivered by *Salmonella*. *Salmonella* T6SS effector Tae4 induces bacterial lysis by cleaving the γ -D-glutamyl-L-meso-diaminopimelic acid amide bond of peptidoglycan and is toxic when expressed in *E. coli* (Russell et al., Cell Host Microbe. 2012, 11:538-549; Benz et al., PLoS One. 2013, 8:e67362). Thus, *E. coli* cells can be lysed by *S. Typhimurium*. We used DH5a as prey following previous methods (Sana et al., Proc. Natl. Acad. Sci. U S A. 2016, 13:E5044-E5051; Song et al., Nat Commun. 2021, 12:423). Nevertheless, we have refocused the paper on the main finding that c-di-GMP directly binds to H-NS and interferes with H-NS binding to DNA in *Salmonella*, and interbacterial competition assays have been removed from the revised manuscript.

The authors show that *V. parahaemolyticus*, *P. aeruginosa* and *P. putida* also bind c-di-GMP, establishing this interaction as a general regulatory principle. Quite unexpectedly, however, the binding sites are not at all conserved, even if their importance is supported by experimental data. I think that more discussion of this point is warranted and perhaps analogous cases could be brought up. In general, given this diversity, I found the analysis of “other” H-NS silencers to be more distracting than helpful because it raises a long list of questions.

Response: We thank the reviewer for this very insightful point. As pointed out by the reviewer, the analysis of interactions of c-di-GMP with H-NS-like proteins in other bacterial species is distracting and, in some cases, not as convincing as the main finding that c-di-GMP modulates DNA binding activity of H-NS in *Salmonella*. Thus, the content with respect to interactions of c-di-GMP with other H-NS proteins in other bacterial species has been removed from the revised manuscript.

Reviewer #2 (Remarks to the Author):

This paper reports the important finding that c-di-GMP is a ligand that

interferes with the H-NS protein's ability to bind DNA. This discovery has implications for our understanding of the very many bacterial systems where H-NS-like nucleoid-associated proteins act to silence transcription. We must now take into account the possibility that the c-di-GMP second messenger is influencing their expression via modulation of the DNA binding activity of H-NS.

Response: We thank the reviewer for the supportive and constructive comments.

As written, the key message of the paper concerning H-NS modulation by direct interaction with c-di-GMP is submerged in a detailed story about type 6 secretion system (T6SS) gene regulation in *Salmonella*. Because far more investigators study H-NS than study *Salmonella* T6SS, the impact of the authors' findings may be diminished if the report retains its current structure and title. Perhaps the authors could consider amending both to give more emphasis to the c-di-GMP and H-NS aspect? One might even be tempted to turn this report into two papers, one for a general journal such as *Nature Comms* on c-di-GMP binding to H-NS, and a separate paper for a microbial pathogenesis journal on T6SS regulation.

Response: We thank the reviewer for the insightful comments and constructive suggestions. As suggested, we have refocused the paper on the main finding that c-di-GMP directly binds to H-NS and modulates DNA binding activity of H-NS in *Salmonella*. Given that c-di-GMP binding to H-NS has been revealed based on c-di-GMP-mediated T6SS regulation, the content with respect to T6SS regulation is retained in the first part of the results.

Throughout the study, the experiments have been executed with great thoroughness. Appropriate controls are included, together with statements of reproducibility and, where required, statistical analysis of the data is provided. The authors place their findings in physiological context and consider their implications for our knowledge of *Salmonella* virulence and of competition with

other gut microbes. However, there remain some surprising gaps in the investigation and the authors should be encouraged to address these prior to publication.

Response: We are very grateful for the positive feedback provided by the reviewer.

One gap concerns the mechanism by which c-di-GMP affects H-NS activity. The paper contains copious data showing that the ligand interferes with H-NS binding to DNA. However, the reader will also want to know if c-di-GMP can displace H-NS from DNA. This requires experiments with H-NS prebound to DNA targets that are treated with increasing concentrations of c-di-GMP. The point is important because we currently have two broad categories of interference with H-NS binding to DNA. One interference mechanism involves displacement of H-NS from DNA, usually by another DNA binding protein (PMID: 18757787) while the other mechanism involves inhibition of H-NS DNA binding activity while the H-NS protein is DNA-free. The Lon protease operates in this second way in *Salmonella* (PMID: 32209674). Could the Lon protease and c-di-GMP act to upregulate H-NS-silenced genes by contributing two independent modes of H-NS inhibition while the protein is off the DNA?. Both of these mechanisms have the effect of reducing the pool of DNA-binding competent H-NS in the cell, increasing the probability that silenced genes will become transcribed. (They are distinct from mechanisms in which H-NS interaction with proteins of the Hha family influences DNA target selection by H-NS.) Some new EMSA experiments will allow the authors to test the c-di-GMP-dependent H-NS displacement hypothesis.

Response: We thank the reviewer for the insightful comments and constructive suggestions. As suggested, we have performed new competitive EMSAs by first incubating H-NS with DNA targets and then adding increasing concentrations of c-di-GMP. Our results showed that c-di-GMP has no ability to displace H-NS from DNA (**Supplementary Fig. 12**). Thus, as pointed out by

the reviewer, the Lon protease and c-di-GMP act to upregulate H-NS-silenced genes by contributing two independent modes of H-NS inhibition while the protein is off the DNA.

A second gap arises because the authors do not seem to address the presence in *Salmonella* of StpA, a closely-related paralogue of H-NS, that is encoded by the chromosomally-located gene, *stpA*. Given the wide sweep of H-NS-like proteins that the authors have considered in their analysis, it seems strange that they have not mentioned this second H-NS-like protein in *Salmonella*. Of the five amino acids in *Salmonella* H-NS implicated in c-di-GMP binding that are highlighted in yellow in Fig 8c, two are not present in StpA. The tyrosine is replaced by phenylalanine and the valine by threonine. In the following alignment, the five amino acids are bracketed by spaces on either side to facilitate identification:

H-NS(St) 91-AARPAKYS Y V D ENGET K TWTGQGR T-115

StpA(St) 91-QPRPAKYR F T D FNGEE K TWTGQGR T-115

Lysine K107 is present in both proteins.

Response: We thank the reviewer for pointing this out. StpA possesses three of the four conserved residues Y99, D101, K107 and T115 in H-NS that are important for c-di-GMP binding (**Supplementary Fig. 10a**). The tyrosine (Y99) in H-NS is replaced by phenylalanine (F98) in StpA. Although both proteins contains the key lysine residue (K106 in StpA corresponding to K107 in H-NS), Y99 of H-NS is also very important for c-di-GMP binding (**Fig. 2f**). According to the model of H-NS_{Ctd} in complex with c-di-GMP (**Fig. 2e**), Y99 forms two hydrogen bonds with c-di-GMP through the hydroxyl group that is absent in phenylalanine. We thus speculate that StpA has low affinity for c-di-GMP. As expected, StpA showed very low binding affinity for c-di-GMP (**Supplementary Fig. 10b**), and mutation of the non-conserved residue to conserved residue within StpA (F98Y) increased its c-di-GMP binding affinity to a level ($0.64 \pm 0.09 \mu\text{M}$) (**Supplementary Fig. 10b**) comparable to that of

H-NS (**Fig. 2b**).

Given the ability of StpA to form heterodimers with H-NS, readers will be curious to know if c-di-GMP can influence the binding of StpA to DNA, either as an StpA homodimer or as a heterodimer with H-NS. The heterodimer situation becomes even more complicated in *Salmonella* clinical isolates that have large plasmids expressing additional H-NS-like proteins (PMID: 20444106).

Response: We thank the reviewer for this insightful comment. In contrast to H-NS, StpA showed very low binding affinity for c-di-GMP (**Supplementary Fig. 10b**). While the intracellular c-di-GMP concentrations in *S. Typhimurium* have been shown to vary from tens of nanomolar to low micromolar levels (Mills et al., *Sci Signal.* 2015, 8:ra57; Petersen et al., *Proc. Natl. Acad. Sci. U S A.* 2019, 116:6335-6340), the K_d values presented here suggest that c-di-GMP is a physiologically relevant ligand for H-NS, but not StpA. Therefore, c-di-GMP obviously can not influence the binding of the StpA homodimers to DNA. When StpA forms heterodimers with H-NS, the interference with H-NS binding to DNA by c-di-GMP may result in a failure of the StpA-H-NS heteromers to participate in DNA-protein-DNA bridging, thus altering the structure of the heteromeric protein-DNA complexes and affecting expression of genes controlled by the StpA-H-NS heterodimers. As suggested by the editor, we do not add new data on the potential roles of StpA, but discuss this point in the revised manuscript (**Lines 347-357**).

There is considerable overlap between the regulons of genes controlled by StpA and H-NS and StpA has a key role in controlling the production of RpoS, a sigma factor whose gene can acquire compensatory mutations in *hns* knockout mutants (PMID: 19843227, PMID: 26631971). Indeed, the *stpA* gene is also prone to the acquisition of compensatory lesions when *hns* is disrupted (PMID: 25375226). Before branching out to a discussion of H-NS-like proteins in other bacterial species, perhaps the authors could make an assessment of

StpA in Salmonella?

Response: We thank the reviewer for the insightful comments and constructive suggestions. Although the previous study (PMID: 25375226) showed that the *stpA* gene is prone to the acquisition of compensatory lesions when *hns* is disrupted, no mutations in *stpA* were detected in the Δhns mutant. However, mRNA level of *stpA* was upregulated more than 2-fold in Δhns compared to the wild type (**Supplementary Fig. 6**), which may partially compensate for the loss of H-NS in regulons of genes controlled by both StpA and H-NS. Furthermore, we found that StpA showed very low binding affinity for c-di-GMP (**Supplementary Fig. 10b**), suggesting that c-di-GMP is not a physiologically relevant ligand for StpA. As suggested by the editor, we have refocused the paper on the main finding that c-di-GMP directly binds to H-NS and modulates DNA binding activity of H-NS in *Salmonella*, and the content with respect to interactions of c-di-GMP with other H-NS-like proteins in other bacterial species has been removed from the revised manuscript. c-di-GMP-mediated regulation of genes controlled by both StpA and H-NS, as well as the StpA-H-NS heterodimers will be further studied in future work.

Lines 168-170: In *Salmonella*, the Fur protein is a negative regulator of *hns* transcription (PMID: 21075923, PMID: 22017966). The authors will need to adjust their description of the roles of the Fur and H-NS proteins in the control of T6SS gene expression to take into account this Fur-*hns* direct regulatory relationship.

Response: We thank the reviewer for this important point. As suggested, we have adjusted the description of the roles of the Fur and H-NS proteins in the control of T6SS gene expression to take into account the Fur-*hns* direct regulatory relationship (**Lines 116-118**).

Lines 179-182: "Despite the lethality of the *hns* null mutation in *S. Typhimurium* strain ATCC 14028s, *S. Typhimurium* strain LT2 has been shown to tolerate

the *hns* mutation (15,26). Consistently, an *hns* deletion mutation was successfully constructed by the CRISPR-Cas9 system in strain SL1344.” Knockout mutations in the *Salmonella* *hns* gene are tolerated when compensatory mutations arise either in the *rpoS* gene or the *phoPQ* operon. Strain LT2 is already defective in expressing full levels of RpoS (the stress and stationary phase sigma factor of RNA polymerase) so it tolerates inactivation of *hns* because, genetically, it is ‘pre-compensated’. This is not the case with strain SL1344, used by the authors of the present study. Here, it is likely that the authors have isolated an SL1344 *hns* mutant that has compensatory lesions in either the *phoPQ* operon or the *rpoS* gene. This should be investigated because, if compensatory mutations are present, their data interpretation must be expanded to take into account the effects of these additional mutations. Both RpoS and PhoPQ exert widespread influence on the expression of *Salmonella* virulence genes, including many of the genes that the authors study in the present investigation.

Response: We thank the reviewer for the insightful comments and constructive suggestions. As suggested, we have examined whether the SL1344 *hns* mutant has compensatory lesions in either the *phoPQ* operon or the *rpoS* gene. However, neither mutations nor changes in the transcription levels of *rpoS* or *phoPQ* were detected in the Δhns mutant compared to the wild type SL1344 (**Supplementary Fig. 6**).

It should perhaps be mentioned that the K107A mutant H-NS can still be displaced from DNA by other mechanisms (e.g. the intervention of anti-repressor proteins). These mechanisms have been described in numerous bacterial systems (PMID: 18757787).

Response: We thank the reviewer for this important point. As suggested, we have discussed the anti-H-NS anti-silencing mechanisms described in the review article (PMID: 18757787) and pointed out that the K107A mutant H-NS can still be displaced from DNA by DNA-binding anti-silencing proteins (**Lines**

344-346).

Minor item

Line 18, Gram-negative, not 'gram-negative' (Gram is a person's name – Hans Christian Gram).

Response: We thank the reviewer for raising this important question. 'gram-negative' has been corrected as 'Gram-negative' in the revised manuscript.

Reviewer #3 (Remarks to the Author):

This paper looks at the connection between the signaling molecule c-di-GMP and the DNA-binding, silencing factor H-NS in Salmonella. The authors argue that (i) c-di-GMP directly binds H-NS, preventing H-NS from binding DNA, (ii) high c-di-GMP levels lead to de-repression of H-NS-silenced genes, (iii) de-repression of H-NS-silenced genes provides a fitness advantage because of increased Type VI secretion, and (iv) the functionally analogous MvaT protein from Pseudomonas is also bound and inhibited by c-di-GMP. These conclusions have major implications for our understanding of the role of c-di-GMP, and for how H-NS activity is modulated. The work is generally very thorough, and I have only one suggested experiment that would add an important control:

Response: The authors are very grateful for the positive feedback provided by the reviewer.

- The authors should use CHIP-qPCR to test whether K107A H-NS binding is affected either by bile salts (or by a more direct modulation of c-di-GMP levels).

Response: We thank the reviewer for this very insightful point. As suggested, CHIP-qPCR has been used to test whether K107A H-NS binding is affected either by bile salts or by overexpression of *adrA* (Fig. 4h, i).

Additional comments:

1. I recommend moving Supplementary Figure 1 into Figure 1.

Response: We thank the reviewer for this important point. As L-arginine and bile salts have been reported to induce c-di-GMP production in *Salmonella* in previous studies (Li et al., Nat Commun. 2022, 13:6684; Mills et al., Sci Signal. 2015, 8:ra57), Supplementary Figure 1 was used to confirm the previous findings. While there are already nine graphs in Figure 1 in the revised manuscript, we keep Supplementary Figure 1 in the Supplementary Information.

2. Change “Relative gene expression” labels on rtPCR graphs to “Relative RNA level”.

Response: We thank the reviewer for this important point. As suggested, “Relative gene expression” labels on rtPCR graphs have been changed to “Relative RNA level”.

3. Data from Supplementary Figures 11 and 13 showing that the K107A H-NS mutant is unaffected by c-di-GMP are key to the main conclusions of the paper. I recommend moving these to the main figures.

Response: We thank the reviewer for this insightful comment. As suggested, Supplementary Figures 11 and 13 have been moved to the main figures as **Fig. 4a** and **Fig. 4b-e**, respectively.

4. Many strains of LT2 are avirulent. Can the authors comment on the genotype of the strain they used?

Response: We thank the reviewer for raising this important question. While some strains of *Salmonella* Typhimurium are avirulent, we used *S.* Typhimurium strain SL1344, a highly virulent strain in mice (Haneda et al., Cell Microbiol. 2012, 14:485-499; Li et al., Nat Commun. 2022, 13:6684).

5. Are there RNA-seq data from *Salmonella* or *Escherichia coli* with altered c-di-GMP levels? If so, the authors could use those data as an independent test of their model, since H-NS-repressed genes are well characterized in these species.

Response: We thank the reviewer for the constructive suggestions. As no RNA-seq data from *Salmonella* with altered c-di-GMP levels have been reported, we have performed RNA-seq to investigate whether changes in intracellular c-di-GMP concentration influence the expression of H-NS-repressed genes throughout the entire genome (**Fig. 5a**). We also found no RNA-seq data from *Escherichia coli* with altered c-di-GMP levels.

6. Can the authors comment on why increased c-di-GMP levels in the two tested *Pseudomonas* species has the same phenotype as deleting *mvaT*, despite the fact that MvaU would be expected to contribute to repression? Related to this, the authors should at least mention MvaU.

Response: We thank the reviewer for raising this important question. In the original manuscript, we focused on the *mvaT* gene as T6SS genes have been identified as MvaT target genes in a previous study (Castang et al., Proc Natl Acad Sci U S A, 2008, **105**:18947-18952). As pointed out by the reviewer, given that MvaT and MvaU bind the same chromosomal regions and coregulate the expression of hundreds of target genes, whether MvaU regulates T6SS gene expression in response to c-di-GMP should be investigated. Nevertheless, as suggested by the editor, we have refocused the paper on the main finding that c-di-GMP directly binds to H-NS and interferes with H-NS binding to DNA in *Salmonella*, and the content with respect to interactions of c-di-GMP with other H-NS-like proteins in other bacterial species has been removed from the revised manuscript.

7. Deletion of *hns* is lethal in other *Salmonella* strains and some other species,

and leads to a growth defect in *E. coli*. Can the authors speculate on the impact of raised c-di-GMP levels on bacterial fitness, since their model indicates that high c-di-GMP levels would derepress all H-NS-silenced genes?

Response: We thank the reviewer for this insightful comment. In fact, deletion of *hns* completely abolished the repressive effect of H-NS on its target genes, whereas elevated intracellular levels of c-di-GMP only partially relieved the repressive effect of H-NS. Raised c-di-GMP levels may lead to minor or no detrimental effect on bacterial fitness. Although xenogeneic silencing by H-NS is thought to prevent the inappropriate expression of horizontally acquired genes and thus suppress their detrimental effect on bacterial fitness, bacteria need to partially derepress H-NS-silenced genes in order to benefit from their expression in specific circumstances. For instance, H-NS-silenced T6SS genes need to be derepressed when the bacterium carrying this weapon competes with other bacteria for survival. H-NS-silenced virulence genes need to be derepressed when bacterial pathogens enter the host environment. Thus, environmental cue-induced elevated intracellular c-di-GMP levels will alleviate the H-NS-dependent repression of gene expression when required. When the environment change, reduced c-di-GMP levels will allow for rapid recovery of the repressive activity of H-NS.

Reviewer #4 (Remarks to the Author):

c-di-GMP is a widespread secondary messenger in bacteria, which regulates numerous different processes, e.g. related to the planktonic/biofilm life style, motility, pathogenicity and more. Over the last decades a number of different effector proteins have been identified that bind and thereby respond to c-di-GMP. In this study, the authors convincingly demonstrate that the highly conserved histone-like nucleoid structuring protein H-NS is, in fact, a c-di-GMP-binding regulator in *Salmonella Enterica*. The authors present in vitro and in vivo approaches that demonstrate direct c-di-GMP binding to H-NS and provide evidence that the c-di-GMP-binding site partly overlaps with the

DNA-binding site of H-NS, so that DNA interaction of the protein is abrogated upon rising levels of c-di-GMP. In *S. Typhimurium* they show by RNAseq approaches that a number of factors are regulated by this mechanism, among them activation of a type VI secretion system (T6SS), which occurs upon exposure of the cells to signals such as bile acids. By mouse models they show that, in fact, the signaling cascade from bile acids to T6SS activation likely functions via c-di-GMP and H-NS and provides an advantage in successful colonization of the host gut. They also present evidence that H-NS in other gammaproteobacteria such as *Vibrio parahaemolyticus*, *Pseudomonas aeruginosa* and *P. putida* binds and responds to c-di-GMP binding, so that this mechanism is likely highly conserved.

Response: We would like to thank the reviewer for careful and thorough reading of this manuscript and for the comprehensive summary of our work.

The major and highly relevant finding that highly conserved H-NS serves as a c-di-GMP-binding regulator is convincing as it is supported by the results presented by the authors. The number of high-quality experiments is impressive and the manuscript is very well structured. I have only some minor comments with respect to this study.

Response: We are very grateful for the positive feedback provided by the reviewer.

58, 'RsmA acts as ...'; this passage does not really fit there and should be moved to somewhere else or could be omitted completely.

Response: We thank the reviewer for this important point. As suggested, the passage has been omitted completely.

153 and throughout manuscript, 'significantly' could also be true to rather small differences. I suggest to at least sometimes mention the factor of changes (e.g. double, six-fold, ..) where appropriate.

Response: We thank the reviewer for raising this important question. As suggested, the factor of changes have been mentioned in several places in the revised manuscript (**Lines 88, 106, 109, 132, 182**).

173, Fig. 2a and throughout manuscript: There are many bar diagrams that provide complex sets of data. To make it a little bit easier for the reader, the authors may consider to add more information to the figure panels, e.g., already indicate in the panel which strain is used.

Response: We thank the reviewer for this important point. As suggested, we have indicated which strain is used in the figure panels of all Figures and Supplementary Figures.

233, please state that alanine substitutions were introduced.

Response: We thank the reviewer for pointing this out. We have added this information in the revised manuscript (**Line 180**).

234, and general, it would be nice to see a presentation of the protein purifications to visualize the quality of the purification. It would also be nice if, for example, a SEC elution profile for the substitution variants could be presented to demonstrate that the structure of the mutated protein is preserved.

Response: We thank the reviewer for this important point. The SDS-PAGE profiles evaluating the quality of the purification for H-NS and its four variants as well as their SEC elution profiles have been presented in **Supplementary Fig. 9**.

259, is FLAG-H-NS functional?

Response: We thank the reviewer for this important point. FLAG-H-NS showed similar levels of DNA- and c-di-GMP-binding activity as the tag-free H-NS (**Supplementary Fig. 13a**), and the expression of the four

H-NS-repressed genes was not altered in FLAG-tagged strain SL1344 compared to untagged SL1344 (**Supplementary Fig. 14**), indicating that the function of H-NS was not impaired by the presence of the FLAG tag.

355, I would ask the authors to add a little bit of explanation here – who is competing and under which conditions.

Response: We thank the reviewer for this important point. Nevertheless, we have refocused the paper on the main finding that c-di-GMP directly binds to H-NS and modulates DNA binding activity of H-NS in *Salmonella*, and the content with respect to interbacterial competition has been removed from the revised manuscript.

389, strongly, better indicate a factor here (see above)

Response: We thank the reviewer for pointing this out. Nevertheless, we have refocused the paper on the main finding that c-di-GMP directly binds to H-NS and modulates DNA binding activity of H-NS in *Salmonella*, and the content with respect to c-di-GMP-mediated activation of the *S. Typhimurium* T6SS within the mouse gut has been removed from the revised manuscript.

390, Fig. S18 shows growth state-dependent, that should be mentioned here.

Response: We thank the reviewer for pointing this out. In fact, we selected a growth phase where the expression of *S. Typhimurium* T6SS genes reach the maximal level when cultured under in vitro conditions (in LB medium to OD₆₀₀ of 3.0 after incubation at 37 °C for 12 h). Nevertheless, the content with respect to c-di-GMP-mediated activation of the *S. Typhimurium* T6SS within the mouse gut has been removed from the revised manuscript.

456, please name the conserved residue

Response: We thank the reviewer for pointing this out. As suggested by the editor and other reviewers, the content with respect to interactions of c-di-GMP

with other H-NS-like proteins in other bacterial species has been removed from the revised manuscript.

Just as a final small remark: in basically all figures, axis labeling or the insets are sometimes really small and hard to make out in the PDF for reviewing.

Response: We thank the reviewer for this important point. We have increased the font size of the labels in all figures.

Reviewer #1 (Remarks to the Author):

This manuscript describes a hitherto unknown regulatory mechanism through direct interactions between two global regulators, H-NS and c-di-GMP. In my opinion, the authors have adequately addressed the concerns/questions raised during the first review cycle and re-focused the manuscript on their most important findings, which have broad implications for our understanding of gene expression control in bacteria.

Reviewer #2 (Remarks to the Author):

The authors have addressed all of the points raised in my review in a satisfactory way.

Reviewer #3 (Remarks to the Author):

The authors have responded well to the reviewer comments. There are several valuable data additions, most notably the StpA data, investigation of whether c-di-GMP can displace DNA-bound H-NS, and the more detailed assessment of the H-NS K107A mutant in Figure 4. Moreover, the manuscript is much easier to read now that the sections on interbacterial competition and H-NS analogues in other species have been removed (these will make for interesting follow-up papers). I think the more focused nature of the paper will increase its impact on the field by making it more accessible.

Minor comments:

- The authors use an LT2 reference genome for their RNA-seq analysis, but the strain they used is SL1344. I recommend reanalyzing the data using the SL1344 reference.
- Another reviewer pointed out that hns is expected to be essential in SL1344, suggesting that there is a suppressor mutation. I suggest whole-genome sequencing of the hns deletion strain.

Reviewer #4 (Remarks to the Author):

In this revised version of the manuscript the authors have adequately addressed my concerns. I have no further remarks.

Response to Reviewers

We wish to begin by thanking the four reviewers for the very supportive and constructive comments. In the revised manuscript, we have addressed the concerns from the reviewers to our best and hope our revision and explanations could answer the reviewer's questions. Please find our detailed responses to each of the comments below.

Reviewer #1 (Remarks to the Author):

This manuscript describes a hitherto unknown regulatory mechanism through direct interactions between two global regulators, H-NS and c-di-GMP. In my opinion, the authors have adequately addressed the concerns/questions raised during the first review cycle and re-focused the manuscript on their most important findings, which have broad implications for our understanding of gene expression control in bacteria.

Response: We would like to thank the reviewer for the very positive comments on our study, and no further issues were raised by the reviewer.

Reviewer #2 (Remarks to the Author):

The authors have addressed all of the points raised in my review in a satisfactory way.

Response: The authors are very grateful for the positive feedback provided by the reviewer. No further issues were raised by the reviewer.

Reviewer #3 (Remarks to the Author):

The authors have responded well to the reviewer comments. There are several valuable data additions, most notably the StpA data, investigation of whether c-di-GMP can displace DNA-bound H-NS, and the more detailed assessment of the H-NS K107A mutant in Figure 4. Moreover, the manuscript is much easier to read now that the sections on interbacterial competition and H-NS analogues in other species have been removed (these will make for interesting

follow-up papers). I think the more focused nature of the paper will increase its impact on the field by making it more accessible.

Response: We would like to thank the reviewer for the very positive comments on our study.

Minor comments:

- The authors use an LT2 reference genome for their RNA-seq analysis, but the strain they used is SL1344. I recommend reanalyzing the data using the SL1344 reference.

Response: We thank the reviewer for pointing this out. While *Salmonella enterica* serovar Typhimurium includes many strains, LT2 is the most widely used strain in genetic studies and many researchers are customary to refer to the gene ID numbers of LT2 when they name genes from other strains such as SL1344 and ATCC 14028 (Mills et al., *Sci Signal*. 2015, 8:ra57; Zheng et al., *Mol Microbiol*. 2013, 89:403-419; Navarre et al., *Science*. 2006, 313:236-238; Li et al., *Nat Commun*. 2022, 13:6684). In fact, the genomic differences between LT2 and SL1344 are very small. While our study does not focus on the very few differential genes between them, an LT2 reference genome used for the RNA-seq analysis of SL1344 are also acceptable.

- Another reviewer pointed out that *hns* is expected to be essential in SL1344, suggesting that there is a suppressor mutation. I suggest whole-genome sequencing of the *hns* deletion strain.

Response: We thank the reviewer for this insightful comment. Previous studies have showed that deletion of *hns* tends to result in compensatory lesions in *stpA*, *rpoS*, or the *phoPQ* operon (Navarre et al., *Science*. 2006, 313:236-238; Ali et al., *PLoS Pathog*. 2014, **10**:e1004500; Lucchini et al., *Mol. Microbiol*. 2009, 74:1169-1186). However, no mutations in these genes were detected in the Δhns mutant of SL1344. We found that mRNA level of *stpA* was upregulated more than 2-fold in Δhns compared to the wild type

(**Supplementary Fig. 6**), which may partially compensate for the loss of H-NS in regulons of genes controlled by both StpA and H-NS. Indeed, as mentioned by the reviewer, we can not eliminate the possibility that a compensatory mutation occurs in other regions of the genome. While our core finding that c-di-GMP inhibits the DNA binding activity of H-NS has been well demonstrated by in vitro and in vivo experiments, whole-genome sequencing of the hns deletion strain will take a lot of time and in the end distract from the main points.

Reviewer #4 (Remarks to the Author):

In this revised version of the manuscript the authors have adequately addressed my concerns. I have no further remarks.

Response: The authors are very grateful for the positive feedback provided by the reviewer. No further issues were raised by the reviewer.